# MRE11 and TREX1 control senescence by coordinating replication stress and interferon signaling

Hervé Técher [1,5], Diyavarshini Gopaul[1,6], Jonathan Heuzé [1], Nail Bouzalmad[1], Baptiste Leray[1], Audrey Vernet[1], Clément Mettling [2], Jérôme Moreaux [2,3,4], Philippe Pasero [1] ✉ & Yea-Lih Lin [1] ✉

Oncogene-induced senescence (OIS) arrests cell proliferation in response to replication stress (RS) induced by oncogenes. OIS depends on the DNA damage response (DDR), but also on the cGAS-STING pathway, which detects cytosolic DNA and induces type I interferons (IFNs). Whether and how RS and IFN responses cooperate to promote OIS remains unknown. Here, we show that the induction of OIS by the H-RAS[V12] oncogene in immortalized human fibroblasts depends on the MRE11 nuclease. Indeed, treatment with the MRE11 inhibitor Mirin prevented RS, micronuclei formation and IFN response induced by RAS[V12]. Overexpression of the cytosolic nuclease TREX1 also prevented OIS. Conversely, overexpression of a dominant negative mutant of TREX1 or treatment with IFN-β was sufficient to induce RS and DNA damage, independent of RAS[V12] induction. These data suggest that the IFN response acts as a positive feedback loop to amplify DDR in OIS through a process regulated by MRE11 and TREX1.

Oncogene-induced senescence (OIS) is a process in which cells stop proliferating due to RS and DNA damage caused by activated oncogenes[1,2]. As such, OIS represents a major barrier to cell transformation. However, how this process is initiated at the molecular level is currently unclear. Two non-mutually exclusive mechanisms have been implicated in OIS. Alterations in the DNA replication program by deregulated oncogenic pathways have been shown to cause DNA damage and induce senescence through the activation of the DDR[3,4]. In addition, recent evidence indicates that OIS also depends on the cyclic GMP-AMP Synthase (cGAS) and the stimulator of interferon genes (STING) pathway[5–7]. Whether the DDR and cGAS-STING pathways cooperate to induce OIS or act independently remains unknown.

Oncogene-induced RS is characterized by alterations in fork progression and the accumulation of DNA damage and chromosome breaks[3,4,8], leading to micronuclei formation[7,9]. One of the best-

characterized examples of OIS results from the overexpression of H-RAS[V12], which constitutively activates the MAPK/ERK pathway. In immortalized human fibroblasts, H-RAS[V12] overexpression has been shown to increase genomic instability by deregulating dNTP pools and by increasing transcription-replication conflicts[4,9–11]. We have recently shown that immortalized human BJ fibroblasts overexpressing H-RAS[V12] (BJ-RAS[V12]) escape senescence by overexpressing factors that promote fork stability, such as Claspin and Timeless[11]. This overexpression has also been observed in primary breast, lung, and colon cancers and is associated with poor prognosis[11]. In BJ-RAS[V12] fibroblasts that escaped senescence, high levels of Claspin and Timeless restored normal fork progression, supporting the view that RS promotes OIS[9,11].

The cytosolic DNA sensor cGAS recognizes double-stranded DNA (dsDNA) fragments longer than 40 bp and produces cyclic guanosine

[1]Institut de Génétique Humaine, University of Montpellier, CNRS, Equipe Labellisée Ligue contre le Cancer, Montpellier, France. [2]Institut de Génétique Humaine, University of Montpellier, CNRS, Montpellier, France. [3]Department of Biological Hematology, CHU Montpellier, Montpellier, France. [4]University of Montpellier, UFR Medicine, Montpellier, France. [5]Present address: Institute for Research on Cancer and Aging of Nice (IRCAN), Université Côte d'Azur, CNRS UMR7284 - INSERM U1081, Nice, France. [6]Present address: Biotech Research and Innovation Centre, University of Copenhagen, 2200 N Copenhagen, Denmark. ✉e-mail: philippe.pasero@igh.cnrs.fr; yea-lih.lin@igh.cnrs.fr

monophosphate–adenosine monophosphate (cGAMP) as a second messenger that activates STING[12]. The STING-TBK1-IRF3 axis then activates the transcription of type-I interferon genes (IFN-α and IFN-β), which subsequently promote the expression of interferon-stimulated genes (ISGs) and other pro-inflammatory cytokines acting as immunomodulators[12,13]. The secretion of cytokines and chemokines is also associated with senescence through a process known as the senescence-associated secretory phenotype (SASP)[14].

The mechanism by which oncogene-induced RS could activate the cGAS-STING pathway in OIS is currently unclear, but cells over-expressing oncogenes display an increased frequency of chromosome breaks and mitotic failures[15], which leads to the formation of micro-nuclei, a major source of cytosolic DNA[7,9]. Since cGAS binds to DNA inside ruptured micronuclei, these structures could promote cGAS-mediated senescence and SASP[6,7].

The link between RS and inflammation may lie in the mechanisms that maintain, repair and signal stalled replication forks. These mechanisms depend on the processing of nascent DNA by a variety of DNA helicases and nucleases that drive the reversion of stalled forks and the controlled degradation of nascent DNA to promote fork restart[16]. MRE11, the catalytic subunit of the MRE11-RAD50-NBS1 (MRN) complex, initiates fork resection through its exo- and endonuclease activities and promotes fork restart in several ways[17–20]. Interestingly, the processing of stalled forks by MRE11 has also been proposed to activate the cGAS-STING pathways through the accumulation of cytosolic DNA species in response to RS[21], providing a potential link between RS and inflammation. Another key factor regulating the cGAS-STING pathway in human cells, TREX1 3′–5′ exonuclease, degrades cytosolic DNA to prevent unscheduled activation of the IFN response[22,23]. In senescent cells, overexpression of the major cytosolic 3′-exonuclease TREX1 was shown to decrease SASP levels during OIS[24], suggesting that cytosolic DNA determines the SASP response. Collectively, these data suggest that the nucleases MRE11 and TREX1 may play important and opposing roles in the induction of senescence in response to oncogene-induced RS. However, this interplay remains to be experimentally addressed.

Here, we overexpressed the H-RAS[V12] oncogene in immortalized human cell lines to investigate the molecular links between oncogene-induced RS and cGAS-mediated DNA sensing in the establishment of OIS. Our results show that the nucleases MRE11 and TREX1 control OIS by modulating RS and cytosolic DNA sensing. Our work reveals a crosstalk between the canonical RS response and the cGAS-STING pathway at the onset of OIS, which has important implications for cancer biology.

## Results

### RAS-induced replication stress correlates with the induction of IFN, ISG and SASP genes

To determine whether and how the RS response and the cGAS-STING pathway cooperate to induce OIS, we generated oncogenic stress in telomerase-immortalized human BJ fibroblasts by overexpressing H-RAS[V12] under the control of a doxycycline-inducible promoter[11]. In these cells, hereafter referred to as BJ-RAS[V12] cells, high levels of RAS[V12] oncogene slowed down replication forks (Fig. 1a). This slowing of fork velocity is the primary manifestation of oncogene-induced RS and is consistent with the results of previous studies[9,11]. RAS[V12] also blocked cell proliferation, an early event in OIS, as evidenced by the sharp reduction in the proportion of BrdU-positive cells upon doxycycline addition (Fig. 1b). The same phenotypes were observed in human IMR90 lung fibroblasts harboring H-RAS[V12] fused to estrogen receptor (IMR90-ER/RAS[V12])[25]. Induction of RAS[V12] by 4-hydroxytamoxifen (4-OHT) led to replication fork slowdown (Supplementary Fig. 1a, b), inhibition of EdU incorporation (Supplementary Fig. 1c, d), formation of senescence-associated heterochromatin foci (SAHF; Supplementary Fig. 1e).

To confirm that the cGAS-STING pathway is induced during OIS in BJ-RAS[V12] cells, we monitored the differential expression of IFN, ISG and SASP genes before and after RAS[V12] induction using RNA-seq[11]. Volcano plots showed increased expression of IFN, ISG and SASP genes in BJ-RAS[V12] cells compared to control BJ cells (Fig. 1c), which is consistent with earlier studies[5–7].

We recently showed that a small fraction of BJ cells maintained under long-term RAS[V12] induction eventually escaped senescence. Interestingly, most of the cells that spontaneously escaped senescence overexpressed factors involved in the maintenance of fork stability, including Claspin and Timeless, which allowed them to restore normal fork progression[11]. To determine whether this adaptation to RS also reduced the expression of IFN, ISG and SASP genes, we analyzed differential gene expression using RNA-seq in two representative clones overexpressing Claspin and Timeless (clones #4 and #5). Remarkably, these two clones showed a marked reduction in IFN, ISG and SASP gene expression compared with senescent BJ-RAS[V12] fibroblasts (Fig. 1c and Supplementary Fig. 1f). In contrast, the expression of IFN and ISG genes did not change relative to BJ-RAS[V12] in a clone that escaped OIS by an uncharacterized mechanism independent of Claspin and Timeless overexpression[11] (Clone #8, Supplementary Fig. 1f). Together, these results suggest that activation of the cGAS-STING pathway in BJ-RAS[V12] fibroblasts correlates with oncogene-induced RS.

### The cGAS-STING pathway promotes RS and senescence upon RAS[V12] induction

To further characterize the interplay between RS and cGAS-STING in OIS, we evaluated the effect of cGAS and STING inhibitors on cell proliferation, fork slowing and SAHF. To this end, BJ-RAS[V12] cells were treated or not with small molecule inhibitors targeting either cGAS (RU.521) or STING (H-151)[26–28] and BrdU incorporation was detected by immunofluorescence (Supplementary Fig. 2a). As described in Fig. 1b, RAS[V12] overexpression induced a 4-fold reduction in the frequency of BrdU-positive cells compared to non-induced cells. However, this effect was strongly reduced when BJ-RAS[V12] cells were treated with the cGAS inhibitor RU.521, and to a lesser extent after STING inhibition (Fig. 2a). Using EdU Click chemistry and flow cytometry to monitor DNA synthesis, we also observed that both cGAS and STING inhibitors used at a lower concentration were sufficient to alleviate the proliferation inhibition induced by RAS[V12] in BJ fibroblasts (Supplementary Fig. 2b). We also investigated whether the cGAS-STING pathway contributes to oncogene-induced replication fork slowdown. Notably, we found that both cGAS and STING inhibitors at least partially rescued fork slowing in BJ-RAS[V12] cells (Fig. 2b). Moreover, cGAS and STING inhibitors reduced SAHF in IMR90-ER/RAS[V12] cells (Fig. 2c). Collectively, these results indicate that oncogene-induced RS and the cGAS-STING pathway are functionally linked to trigger OIS in BJ and IMR90 fibroblasts.

### MRE11 is required for OIS in BJ-RAS[V12] cells

We have recently reported that MRE11 acts at the interface between RS and the cGAS-STING pathway[21]. To determine whether MRE11 is also involved in OIS, we next compared the growth of BJ cells over-expressing RAS[V12] in the presence or the absence of the MRE11 inhibitor Mirin[29]. Proliferation was strongly inhibited in cells overexpressing RAS[V12], but was fully restored in the presence of 10 μM Mirin (Fig. 3a), indicating that MRE11 plays an important role in the growth arrest induced by RAS[V12].

To confirm that the enzymatic activity of MRE11 promotes OIS, we next analyzed the effect of Mirin on senescence-associated β-galacto-sidase (SA-β-gal), another classical hallmark of senescent cells. The percentage of BJ-RAS[V12] fibroblasts exhibiting β-galactosidase activity increased significantly on days 8 and 14 after H-RAS[V12] induction. However, SA-β-gal activity was no longer detected after Mirin treatment (Fig. 3b, c). We also analyzed the role of MRE11 on SAHF

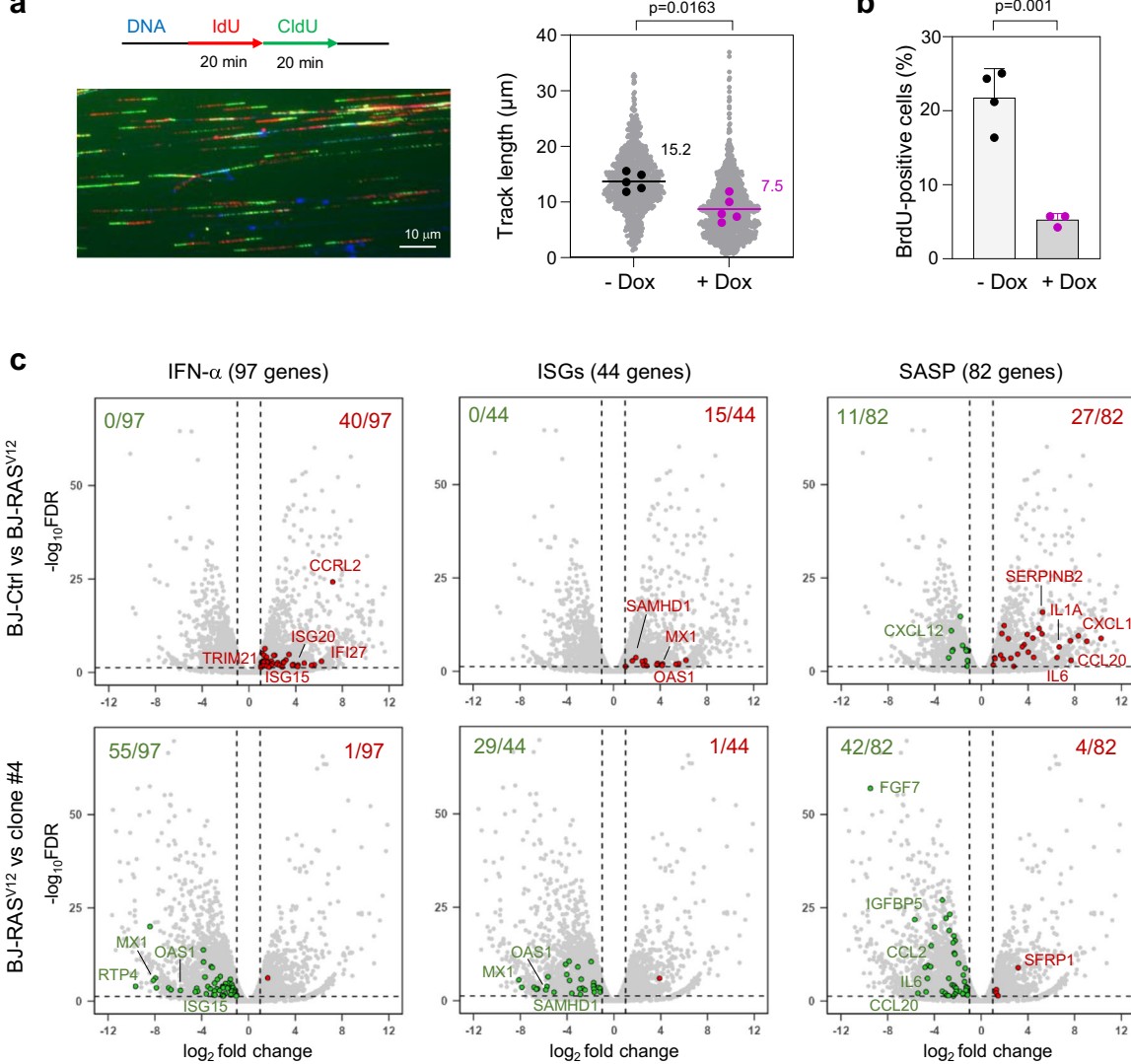

**Fig. 1 | Induction of IFN, ISG and SASP genes in response to RAS^V12-induced RS.**
**a** Analysis of replication fork progression by DNA fiber spreading. Cells were pulse-labeled with two consecutive pulses of IdU and CldU for 20 min each. DNA was counterstained with an anti-ssDNA antibody and monoclonal antibodies against IdU and CldU. A representative image is shown. RAS^V12 overexpression was induced (+Dox) or not (−Dox) with 10 μg/ml doxycycline for 8 days. Gray dots correspond to track lengths from five independent experiments. Track length is the sum of adjacent IdU and CldU tracks from individual forks. The mean of individual experiments is shown as large dots. Horizontal bars indicate the median of the five biological replicates. P value: two-sided paired t-test. **b** RAS^V12 overexpression was induced (+Dox, n = 3) or not (−Dox, n = 4) with doxycycline for 8 days and cells were labeled with 30 μM BrdU for 20 min. BrdU incorporation was monitored by immunofluorescence of at least 100 cells per condition and in each independent experiment. Means and standard deviations are presented for at least three independent experiments. P value: two-sided unpaired t-test. **c** Volcano plots of differentially expressed genes in BJ-RAS^V12 cells overexpressing (BJ-RAS^V12) or not (BJ-Ctrl) RAS^V12. Clone #4 is a BJ-RAS^V12 clone that escaped senescence by overexpressing Claspin and Timeless[11]. Gene sets correspond to interferon-α response genes (IFN-α, GSEA M5911), interferon stimulated genes (ISGs, Reactome R-HSA-9034125) and SASP genes[69]. Up-regulated genes are shown in red (log_2 fold change > 1 and FDR < 0.05). Down-regulated genes are shown in green (log_2 fold change < −1 and FDR < 0.05). Data are from biological duplicates. Source data are provided as a Source Data file.

formation in IMR90-ER/RAS^V12 fibroblasts. Mirin treatment prevented the formation of SAHF induced by RAS in a dose-dependent manner (Supplementary Fig. 3a, b). Mirin also restored BrdU incorporation in BJ-RAS^V12 cells (Fig. 3d and Supplementary Fig. 3c). Together, these data indicate that the catalytic activity of MRE11 is required to promote OIS in RAS^V12-overexpressing cells.

Mirin is a potent inhibitor of the exonuclease activity of MRE11[29], but this enzyme also displays endonuclease activity. To determine which of these two activities is involved in OIS, BJ-RAS^V12 cells were treated with inhibitors targeting the endonuclease (PFM01) or exonuclease (PFM39) activity of MRE11[30]. The fraction of cells that incorporated BrdU after 8 days of H-RAS^V12 induction was restored by PFM01 to the same extent as Mirin, but was not restored as efficiently by

PFM39 (Fig. 3c), suggesting that both endonuclease and exonuclease activities of MRE11 contribute to OIS.

### MRE11 slows down replication forks in BJ-RAS^V12 cells

To further characterize the mechanism by which the catalytic activity of MRE11 mediates RS and OIS under oncogenic stress, we next monitored its effect on fork progression. BJ-RAS^V12 fibroblasts treated with or without 10 μM Mirin were pulse-labeled with IdU and CldU and the length of replicated tracks was measured by DNA fiber spreading. At this concentration, Mirin had only a modest effect on fork speed in the absence of oncogenic stress (Fig. 4a). However, Mirin completely suppressed the fork slowing mediated by RAS^V12 induction when added either immediately upon H-RAS^V12 induction (Fig. 4a) or five days after

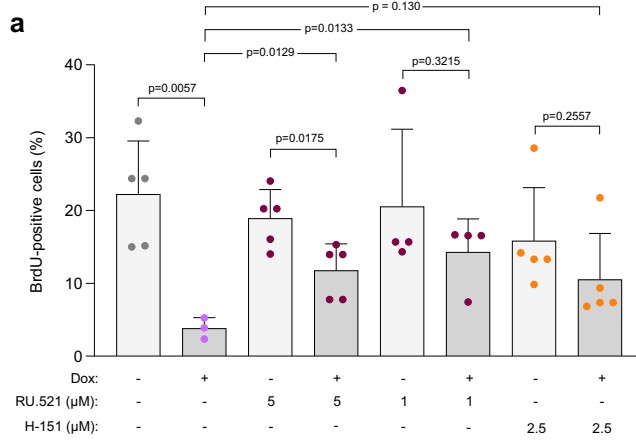

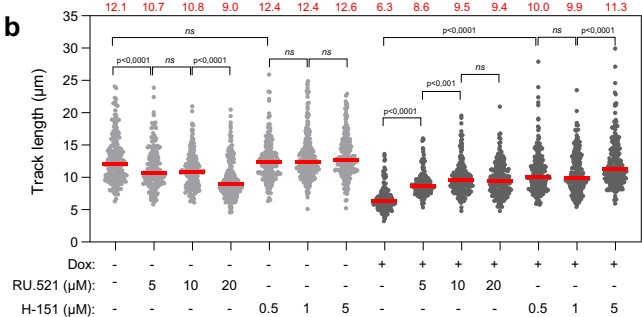

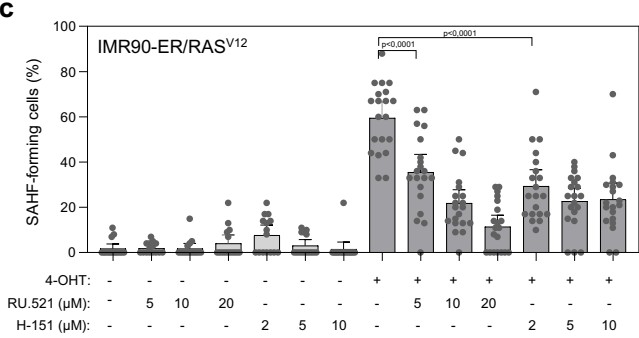

**Fig. 2 | The cGAS-STING pathway contributes to RAS^V12-induced senescence.**
**a** Inhibition of the cGAS-STING pathway restores cell proliferation in BJ-RAS^V12 fibroblasts. RAS^V12 overexpression was induced (+Dox) or not (−Dox) for 8 days with 10 µg/ml doxycycline and cells were labeled for 20 min with 30 µM BrdU. Cells were treated with the cGAS inhibitor RU.521 (1 or 5 µM) or the STING inhibitor H-151 (2.5 µM). DMSO was used as control. BrdU incorporation was monitored by immunofluorescence and the frequency of BrdU-positive cells was determined. At least 62 cells per condition in each experiment were scored. Mean, SD and p values (two-sided unpaired t-test) are shown. Each point represents a biological replicate (n = at least 3). **b** cGAS and STING inhibitors counteract RAS^V12-induced fork slowing. BJ-RAS^V12 fibroblasts were treated or not with 10 µg/ml doxycycline in the absence or presence of increasing doses of cGAS or STING inhibitor (RU.521 or H-151, respectively) for 6 days. Replication fork progression was measured by DNA fiber spreading. Median of the IdU+CldU track length is indicated in red. ****p < 0.0001, ***p < 0.001, ns non-significant, two-tailed Mann–Whitney rank sum test. A minimum of 150 fibers were measured in each sample. A representative experiment is shown (n = 2). **c** cGAS and STING inhibitors prevent SAHF formation induced by RAS^V12. IMR90-ER/RAS^V12 fibroblasts were treated or not with 100 nM 4-hydroxytamoxifen (4-OHT) in the absence or presence of increasing doses of cGAS or STING inhibitor (RU.521 or H-151, respectively) for 6 days. The percentage of SAHF was quantified by DAPI staining. ****p < 0.0001, two-tailed Mann–Whitney rank sum test. One representative experiment from two independent experiments is shown (n = 2). Source data are provided as a Source Data file.

induction (Supplementary Fig. 4a). Similar suppression of fork slowing by Mirin was obtained when oncogenic stress was induced in IMR90-ER/RAS^V12 fibroblasts (Supplementary Fig. 4b). We also observed an increased frequency of asymmetric IdU and CldU track lengths in BJ-RAS^V12 cells, which is indicative of fork arrest[31] and was again reduced by Mirin treatment (Fig. 4b).

Since BrdU incorporation in BJ-RAS^V12 fibroblasts was rescued more efficiently by Mirin and by the endonuclease inhibitor PFM01 than by the exonuclease inhibitor PFM39 (Fig. 3c), we next investigated the impact of these molecules on fork progression with or without RAS^V12 induction (Supplementary Fig. 4c). In the absence of oncogenic stress, MRE11 inhibitors had little to no effect on fork velocity (Fig. 4c). However, both Mirin and PFM01 rescued RAS^V12-induced fork slowing to the same extent, in contrast to the exonuclease inhibitor PFM39 (Fig. 4c). Altogether, these results suggest that the replication stress induced by the overexpression of H-RAS^V12 is mediated by both endonuclease and exonuclease activities of MRE11.

## MRE11 promotes the formation of micronuclei and activates the DDR in BJ-RAS^V12 cells

Micronuclei form under mild RS conditions because of unrepaired DSBs, incomplete DNA replication, and chromosome non-disjunction[32,33]. In BJ-RAS^V12 fibroblasts, the frequency of micronuclei increased by ~10-fold after 5 days of RAS^V12 induction and further increased after 8 days (Fig. 4d, e). Since MRE11 plays an active role in RAS^V12-induced RS, we asked whether it contributed to the formation of micronuclei. To address this possibility, BJ-RAS^V12 fibroblasts were treated or not with 10 µM Mirin in the presence or absence of RAS^V12 induction. Mirin treatment did not affect the percentage of micronucleated cells in the absence of oncogenic stress, but dramatically reduced it after 5 and 8 days of H-RAS^V12 induction (Fig. 4d, e), indicating that MRE11 plays an active role in the formation of RAS-induced micronuclei. RAS^V12 induction also activated the DNA damage response, as shown by increased levels of phospho-ATM (S1981) and phospho-RPA32 (S33; Fig. 4f). Interestingly, this DDR activation was prevented by MRE11 inhibition (Fig. 4f), which is consistent with earlier studies[29] and with the effect of Mirin on micronuclei formation.

## MRE11 promotes the induction of IFN, ISG and SASP genes

Micronuclei represent a major source of cytosolic DNA and have been implicated in the activation of the cGAS-STING pathway through rupture of their nuclear envelope[34–36]. Since MRE11 promotes micronuclei formation in BJ-RAS^V12 cells, we next investigated whether MRE11 is required for the induction of the IFN response in these cells. To address this possibility, the induction of ISG15 and IL-1α was analyzed by immunoblotting in BJ-RAS^V12 fibroblasts, treated or not with Mirin. After RAS^V12 induction, ISG15 and IL-1α levels increased, but were reduced in the presence of Mirin, to the same extent as phospho-ATM and phospho-RPA (Fig. 4f and Supplementary Fig. 5). To confirm these results, we next analyzed the differential expression of IFN, ISG and SASP genes by RNA-seq in BJ-RAS^V12 fibroblasts treated or not with 10 µM Mirin. This analysis revealed a reduction in IFN, ISG and SASP gene expression upon Mirin treatment (Fig. 5a), which was further confirmed by RT-qPCR for IL6 and CXCL1 genes (Fig. 5b). Cell fractionation and Western blotting also confirmed that Mirin did not affect the distribution of MRE11 and cGAS in cytoplasmic and nuclear compartment (Supplementary Fig. 5). Altogether, these data indicate that MRE11 contributes to the activation of IFN response in BJ-RAS^V12 fibroblasts, which in turn promotes OIS.

## TREX1 activity modulates OIS

To assess the contribution of micronuclei to the onset of senescence, we examined the effect on OIS of overexpressing the major cytosolic nuclease TREX1, which degrades cytosolic DNA. To this end, we constructed BJ-RAS^V12 cells constitutively overexpressing GFP-tagged

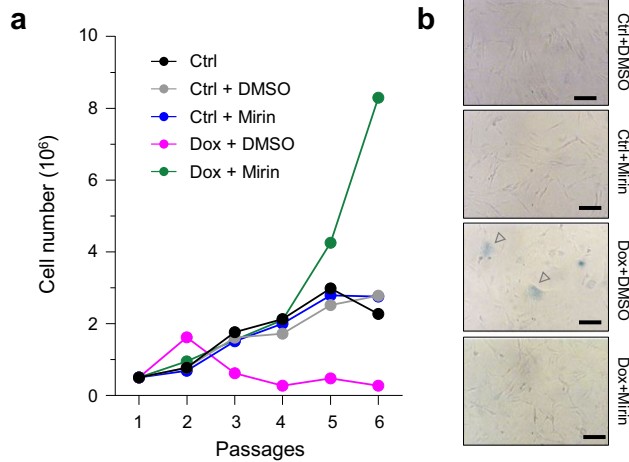

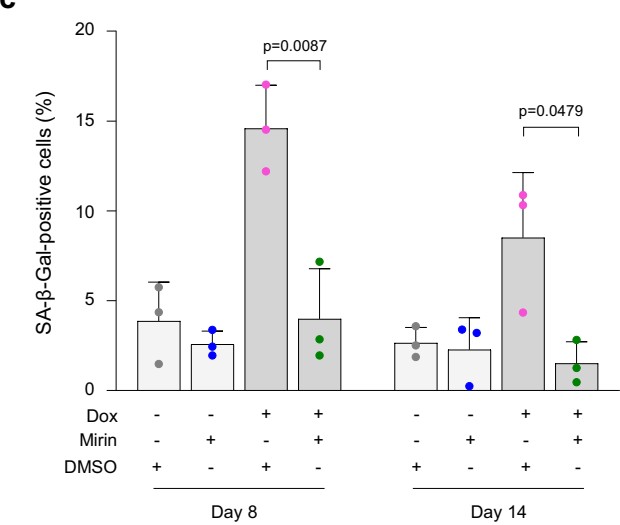

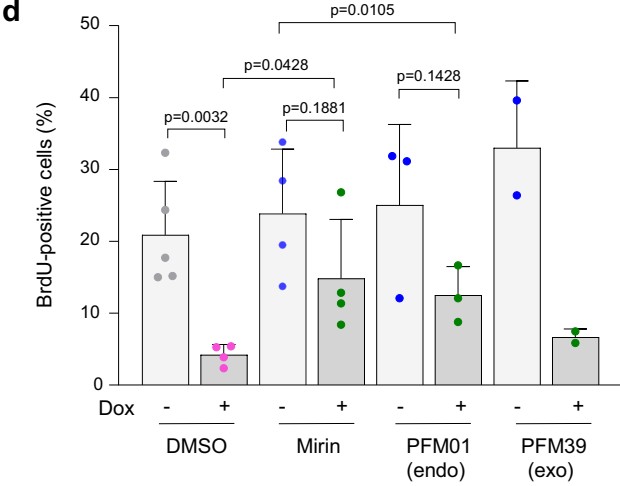

**Fig. 3 | MRE11 promotes senescence in BJ-RAS^V12 fibroblasts. a** Effect of MRE11 inhibition on BJ-RAS^V12 cell growth. Untreated (Ctrl) or RAS-induced (Dox, 10 µg/ml) BJ-RAS^V12 cells were grown in the presence (+Mirin) or the absence (DMSO) of 10 µM Mirin. Cumulative cell number was calculated by counting cells during five consecutive passages, with each passage lasting 3–4 days in culture. A representative experiment is shown (n = 2). **b** Representative images of SA-β-gal staining in BJ-RAS^V12 fibroblasts overexpressing RAS^V12 and treated or not with 10 µM Mirin for 8 and 14 days (n = 3). Scale bars are 100 µm. **c** MRE11 inhibition prevents the induction of senescence-associated β-galactosidase (SA-β-gal). BJ-RAS^V12 fibroblasts were induced with 10 µg/ml doxycycline (Dox) for 8 and 14 days in the presence or absence of 10 µM Mirin. Cells were stained for SA-β-gal activity and the frequency of SA-β-gal positive cells was scored. Mean, SD and p values (two-sided unpaired t-test) are shown for three independent experiments. **d** Effect of MRE11 inhibitors on cell proliferation. BJ-RAS^V12 fibroblasts were induced or not with 10 µg/ml doxycycline (Dox). The frequency of BrdU-positive cells, at least 74 cells were scored per condition in each independent experiment, before and after RAS^V12 induction was monitored in the presence or the absence of 10 µM Mirin, 10 µM PFM01 and 10 µM PFM39 after 8 days of treatment. Non-treated (-) and DMSO-treated cells were used as controls. Mean, SD, and p values (two-sided unpaired t-test) are shown. Each point represents a biological replicate. Source data are provided as a Source Data file.

overexpressing H-RAS^V12 (Fig. 6d). Importantly, TREX1-D18N also induced SA-β-gal production in BJ fibroblasts in the absence of H-RAS^V12 (Fig. 6d), indicating that its dominant-negative effect on endogenous TREX1 is sufficient to trigger senescence. Overexpression of the TREX1-D18N mutant also increased the expression of ISG (Fig. 6e), IFN-α (Supplementary Fig. 6b), *IL6* and *CXCL1* genes (Supplementary Fig. 6c). Together, these results indicate that TREX1 negatively regulates OIS by degrading cytosolic DNA. Conversely, nuclease-dead TREX1 is sufficient to trigger senescence independently of RAS^V12 induction, presumably by allowing the accumulation of endogenous cytosolic DNA and by promoting IFN signaling.

To determine whether high levels of TREX1 or TREX1-D18N affect the speed of replication forks and induce DNA damage, we next generated Doxycycline-inducible constructs (Supplementary Fig. 6d) and monitored fork velocity and DNA damage after 6 days of induction. DNA fiber spreading experiments showed that TREX1 or TREX1-D18N had little effect on fork progression (Supplementary Fig. 6e) and a modest but statistically significant effect on DNA damage, scored here with the comet assay (Supplementary Fig. 6f). These data suggest that TREX1 modulates OIS by controlling cytosolic DNA levels independently of replication stress and DNA damage. However, the frequency of 53BP1 bodies increased after TREX1-D18N overexpression, particularly when combined with RAS^V12 overexpression (Supplementary Fig. 6g). In addition, we detected a proportional increase in micronuclei frequency, a fraction of which were positive for cGAS (Supplementary Fig. 6h). Altogether, these data indicate that TREX1 inhibition does not directly generate RS but synergizes with RAS^V12 to promote the formation of chromosome breaks and micronuclei, triggering cGAS-STING signaling and OIS.

### IFN-β induces replication stress and senescence independently of RAS^V12

Since TREX1 inhibition was sufficient to engage BJ cells into senescence, we next investigated whether this was due to the induction of the IFN response. To analyze the role of type-I IFNs in this process, BJ-RAS^V12 cells were supplemented with recombinant interferon-β (IFN-β) and the fraction of cells entering senescence was assessed using the SA-β-gal assay. Strikingly, IFN-β increased the percentage of SA-β-gal-positive cells by more than three-fold, which was even greater than the effect of RAS^V12 induction alone (Fig. 7a). IFN-β also reduced the frequency of BrdU-positive cells in a dose-dependent manner (Fig. 7b). In addition, DNA fiber spreading experiments revealed a strong reduction in fork velocity after IFN-β treatment, to the same extent as after RAS^V12 induction (Fig. 7c). We also observed a dose-dependent increase

versions of either wild-type (TREX1) or a dominant-negative nuclease-dead (TREX1-D18N) protein[37,38] (Fig. 6a). Both proteins were predominantly located in the cytosol (Fig. 6b) and accumulated within ~35% of micronuclei (Fig. 6c and Supplementary Fig. 6a). Overexpression of GFP-TREX1 reduced the level of RAS-induced SA-β-gal activity compared to that in control cells (Fig. 6d), supporting the view that cytosolic DNA contributes to OIS. In contrast, overexpression of the nuclease-dead mutant TREX1-D18N rather increased OIS in cells

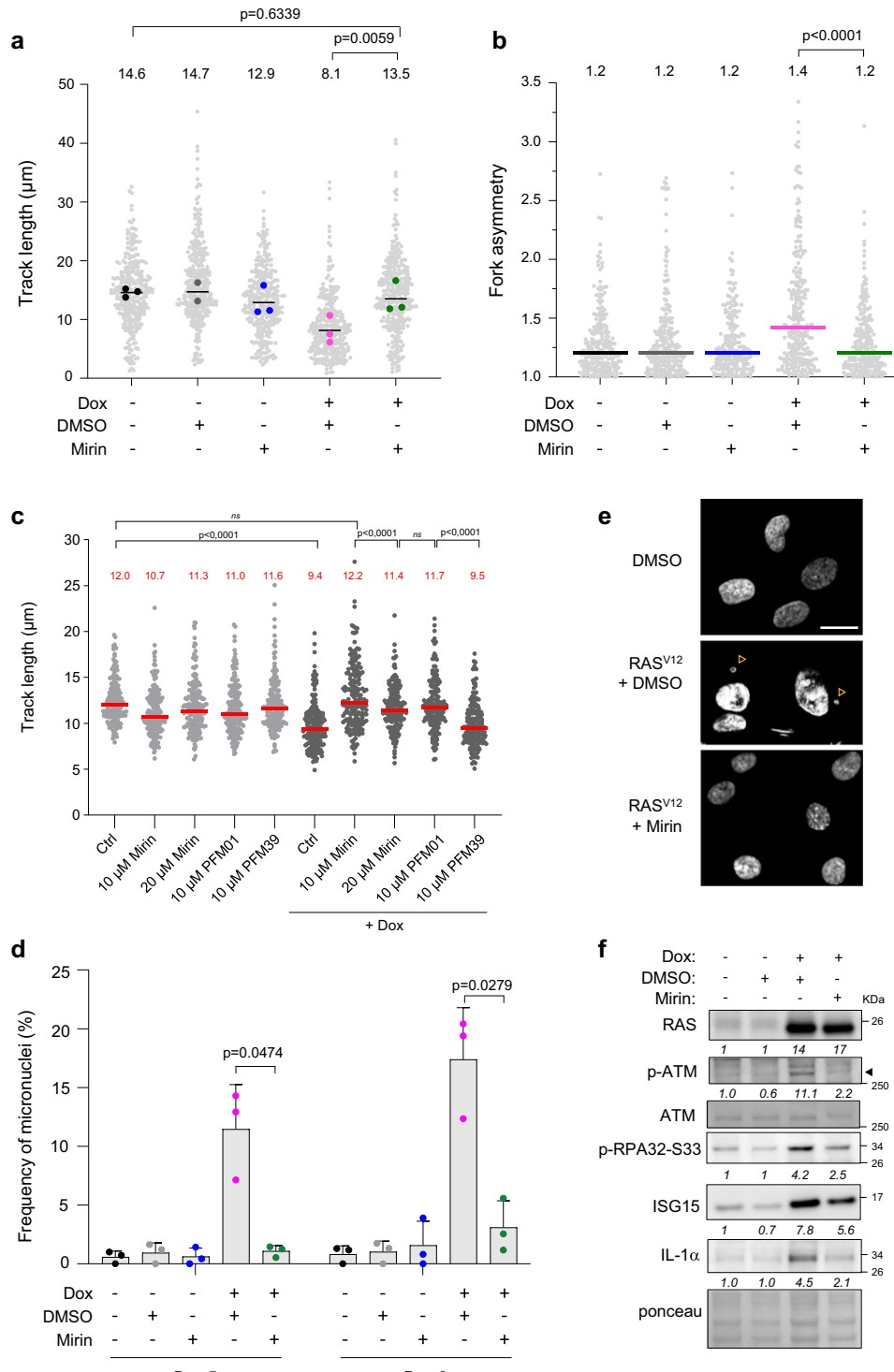

in micronuclei in BJ cells exposed to IFN-β, supporting the view that IFN-β causes RS, and this effect was additive to the effect of RAS^V12 (Fig. 7d). Importantly, we observed the same effect of IFN-β on immortalized human fibroblasts IMR90 and on retinal pigment epithelial RPE-1 cells. In IMR90 cells, IFN-β strongly reduced the fraction of cells incorporating EdU (Supplementary Fig. 7a) and the speed of replication forks (Supplementary Fig. 7b). In RPE-1 cells, it also slowed down growth (Supplementary Fig. 7c) and fork progression (Supplementary Fig. 7d) in a dose-dependent manner. We also observed an IFN-β-dependent accumulation of micronuclei in RPE-1 cells (Supplementary Fig. 7e). In BJ fibroblasts, treatment with IFN-β was sufficient

to induce 53BP1 foci, a DSB marker (Fig. 7e) and this effect was additive with RAS^V12 to form both 53BP1 foci and micronuclei (Fig. 7e and Supplementary Fig. 7f). Next, we used the comet assay to determine whether the IFN-β treatment was sufficient to cause DNA damage in IMR-90 cells. Interestingly, we observed that IFN-β inflicted DNA breaks in a dose-dependent manner (Supplementary Fig. 7g).

To further characterize the interplay between IFN-β and RAS^V12, we next analyzed their combined effect on fork progression using DNA fiber spreading. We found that IFN-β further reduced fork speed (Supplementary Fig. 7h) and SA-β-gal staining (Supplementary Fig. 7i) in BJ cells overexpressing RAS^V12. Notably, incubation of BJ-RAS^V12 cells

**Fig. 4 | Mirin prevents RAS^V12^-induced replication stress. a** BJ-RAS^V12^ cells were grown for 8 days with or without 10 μg/ml Dox and 10 μM Mirin. Cells were labeled with two consecutive pulses of IdU and CldU for 20 min each and DNA fibers were stretched on glass slides. The distribution of track length is shown for at least two independent experiments. A minimum of 300 individual forks were measured per condition. Median and p values (two-sided paired t-test) are indicated. **b** Mirin suppresses the fork asymmetry induced by RAS^V12^. Fork asymmetry was measured as the ratio of the longest to the shortest track for each individual fork in BJ-RAS^V12^ cells treated with different combinations of Dox and Mirin. Median and p values (two-tailed Mann−Whitney rank sum test) are indicated. At least 200 individual forks were measured for each condition. A representative experiment is shown (n = 2). **c** Mirin and PFM01, but not PFM39, restore RAS^V12^-induced fork slowing. BJ-RAS^V12^ fibroblasts were treated or not with 10 μg/ml doxycycline in the absence or presence of increasing doses of Mirin, 10 μM PFM01 or 10 μM PFM39 for 6 days.

Replication fork progression was measured by DNA fiber spreading. A minimum of 150 fibers were measured in each sample. Median track length is indicated in red. ****p < 0.0001, two-tailed Mann−Whitney rank sum test. A representative experiment is shown (n = 2). **d** Mirin prevents the accumulation of micronuclei in BJ-RAS^V12^ fibroblasts. The frequency of micronucleated cells was determined by fluorescence microscopy in cells treated with indicated combinations of 10 μg/ml Dox and 10 μM Mirin. At least 50 cells were analyzed per condition and in each experiment. Mean, SD, and p values (two-sided unpaired t-test) are shown for three independent experiments. **e** Representative images of DAPI-stained BJ-RAS cells after 5 days of RAS^V12^induction. Arrowheads point to micronuclei. Scale bar is 20 μm. **f** Western blot analysis of DDR factors, ISG15 and IL-1α in BJ-RAS^V12^ cells treated or not with 10 μg/ml doxycycline and 10 μM Mirin. Relative fold changes are indicated. The experiment is representative of at least two independent experiments (see also Supplementary Fig. 5). Source data are provided as a Source Data file.

with an antibody against the IFN-α/β receptor partially rescued the RAS^V12^-dependent fork slowing (Fig. 7f), suggesting that the IFN-β produced upon RAS^V12^ induction contributes to a further reduction in fork speed. In addition, the anti-IFN-β receptor also partially rescued RAS-induced senescence (Fig. 7g), supporting the view that oncogene-induce RS and the IFN response cooperate to promote OIS.

## Discussion

It is now well established that OIS can be triggered by two distinct events, RS and IFN signaling[3–7], but whether these two pathways promote OIS independently or are functionally linked is currently unknown. Our results indicate that the nucleases MRE11 and TREX1 modulate both processes to regulate senescence in different immortalized human cell lines overexpressing H-RAS^V12^, providing a mechanistic link between oncogene-induced RS and IFN signaling in OIS.

MRE11 forms a heterotrimeric complex with RAD50 and NBS1 and plays pleiotropic roles in maintaining genome stability, including HR-mediated DSB repair, ATM activation, and replication fork repair[39,40]. As such, the MRN complex represents a barrier to cancer[41–43]. Importantly, MRE11 is also required to activate cytosolic nucleic acid sensing under RS conditions[16,21] and to induce SASP in senescent cells[44,45]. Thus, MRE11 is strategically located at the interface between oncogene-induced RS and IFN signaling. Here, we show that MRE11 links these two pathways in RAS^V12^-induced senescence. Indeed, MRE11 plays a direct role in fork slowing and micronuclei formation in response to RAS^V12^ overexpression in immortalized human fibroblasts. It is also required for the induction of IFN, ISG, and SASP genes, growth inhibition, SAHF formation, and induction of SA-β-gal activity. Thus, MRE11 plays a central role in OIS by linking the RS and IFN responses to oncogenic stress.

One of the earliest manifestations of RAS^V12^-induced RS is a dramatic reduction in fork velocity[4,9,11,46]. Our data show that this fork slowdown is suppressed by Mirin, suggesting that the tumor-suppressive function of MRE11 comes at the cost of slower replication. This fork slowing may represent a general fork protection mechanism, as MRE11 also reduces fork speed in cells exposed to different sources of RS, such as progeria[47], cohesion defects[48] and inhibition of topoisomerase 1 by camptothecin[49]. Interestingly, a recent study indicates that MRE11 slows down elongation in naïve mouse embryonic stem cells as a consequence of increased transcription-replication conflicts[50], which is reminiscent of BJ-RAS^V12^ cells. The mechanism by which MRE11 activity regulates fork speed remains unknown. This may be related to the formation of post-replicative ssDNA gaps[51,52], which activate ATR and may induce a checkpoint-dependent slowdown of DNA synthesis[53–55]. Alternatively, MRE11 could modulate the formation or stability of reversed forks[16,56,57], which could indirectly affect overall replication velocity. Excessive MRE11 activity in cells overexpressing RAS^V12^ could also result in inefficient HR-mediated fork repair, chromosome breaks and micronuclei formation.

MRE11 exhibits both endonuclease and exonuclease activities, with the latter being efficiently repressed by Mirin[29]. In BJ-RAS^V12^ cells, Mirin prevented the onset of senescence hallmarks, suggesting that MRE11 acts through its exonuclease activity to promote OIS. However, we found that PFM01, a specific inhibitor of the endonuclease activity of MRE11, prevented fork slowdown and replication inhibition upon RAS^V12^ induction. In contrast, inhibition of the exonuclease activity of MRE11 with PFM39 did not reverse the effects of RAS^V12^. Together, these data suggest that MRE11 promotes OIS via its endonuclease activity, which could also be repressed by Mirin. Since PFM01 prevents fork cleavage in the absence of RAD51[58], an attractive possibility could be that MRE11 promotes the formation of micronuclei in BJ-RAS^V12^ cells by cleaving stalled forks, which would in turn activate the cGAS-STING pathway[34–36].

In cells exposed to chronic RS, micronuclei assemble after mitosis around lagging chromosomes or large chromosome fragments[32,33]. In BJ-RAS^V12^ cells, micronuclei formation was largely suppressed by Mirin, which is consistent with the view that micronuclei result from oncogene-induced RS. As reported for situations involving RS and DSB induction[5,6,34–36], we observed an accumulation of cGAS in a fraction of the micronuclei in BJ-RAS^V12^ fibroblasts. Although our data indicate that cGAS is required for RAS^V12^-induced senescence, they do not formally demonstrate that cGAS is activated inside these micronuclei, which contain nucleosomes that sequester cGAS and prevent its activation[59–61]. Interestingly, it has been recently reported that the MRN complex binds to nucleosome fragment to displace cGAS from acidic-patch-mediated sequestration and activate it[62]. Unlike its role in OIS, this function of MRE11 in cGAS activation is independent of its nuclease activity. Together, these new findings indicate that MRE11 links DNA damage and cGAS-STING signaling by two independent and complementary mechanisms.

To investigate the contribution of micronuclei and cGAS to OIS, we overexpressed the major cytosolic nuclease TREX1 in BJ-RAS^V12^ fibroblasts. TREX1 was recently shown to counteract cGAS signaling by degrading DNA in micronuclei[35] and preventing the induction of SASP genes in senescent cells[24]. Here, we found that TREX1 overexpression suppressed SA-β-gal activity in BJ-RAS^V12^ cells, supporting the view that RS-mediated micronuclei activate cGAS in OIS. Unexpectedly, we found that the overexpression of a dominant-negative form of TREX1 termed TREX1-D18N was sufficient to induce SA-β-gal activity in the absence of RAS^V12^ induction to the same extent as in cells overexpressing RAS^V12^. In addition, treatment of BJ and RPE-1 cells with IFN-β slowed down fork progression, impeded DNA synthesis and generated DNA breaks and micronuclei in a dose-dependent manner. These results suggest that activation of the IFN pathway is sufficient to promote senescence, independently of oncogene-induced RS. The mechanism by which IFN-β induces RS is currently unclear. It could act via the ubiquitin-like protein ISG15, a major ISG induced by type I IFNs that causes RS and DNA damage when overexpressed[63]. Interestingly, we also found that blockage of the IFN-β receptor in cells

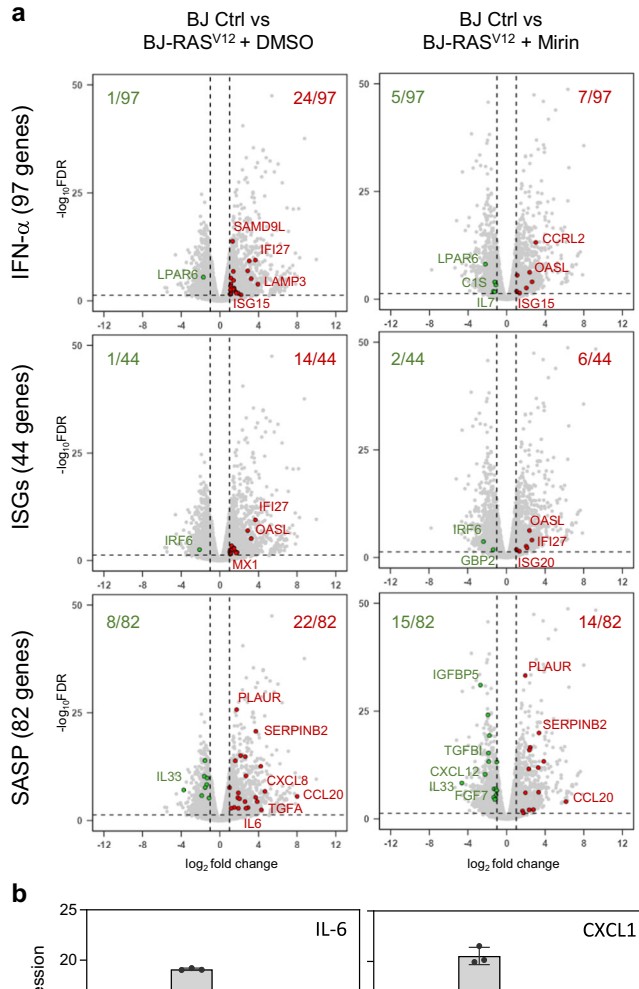

**Fig. 5 | The induction of IFN, ISG and SASP genes in BJ-RAS^V12 cells depends on MRE11. a** BJ-RAS^V12 fibroblasts were induced for 8 days with 10 μg/ml Dox in the presence or the absence of 10 μM Mirin. Volcano plots of differentially expressed IFN, ISG and SASP genes between control and RAS-induced BJ cells treated or not with Mirin are shown. Data are from biological triplicates. **b** Levels of IL6 and CXCL1 expression were quantified by RT-qPCR. Mean ± SD of one representative experiment of two independent experiments is shown (n = 2). Source data are provided as a Source Data file.

overexpressing RAS^V12 partially alleviated fork slow down, to the same extent as cGAS and STING inhibition. These data suggest that the IFN response induced by RS acts as a positive feedback loop to reinforce RS in response to RAS^V12 induction.

Collectively, our results support a model in which the RS and cytosolic DNA sensing pathways do not act independently to induce OIS upon H-RAS^V12 overexpression, but are rather connected through MRE11 (Fig. 8). In this model, micronuclei resulting from replication defects induces cytosolic DNA and activates the cGAS-STING pathway. Importantly, the cGAS-dependent induction of IFNs and ISGs further increases RS, acting as a positive feedback loop that amplifies the RS response. This positive feedback loop explains why the addition of IFN-

β is sufficient to induce RS, independent of RAS^V12. This mechanism is also supported by our observation that replication fork defects induced by RAS^V12 depends, at least in part, on cGAS-STING signaling and IFN-receptors. This response to cytosolic DNA is under the control of TREX1 nuclease, which prevents unscheduled senescence by degrading cytosolic DNA in normal cells.

Our findings raise important questions regarding the mechanisms by which MRE11, TREX1-D18N, and IFN-β induce RS and on the potential role of downstream factors such as ISG15 in this process. Further work is also required to demonstrate that inflammation-mediated RS is necessary and sufficient to recapitulate the effect of RAS^V12 in vivo. Addressing these issues will allow us to better understand the interplay between replication stress and inflammation during oncogene-induced senescence and how this important defense mechanism protects complex organisms from cell transformation.

## Methods
### Cell culture
Normal human fibroblasts BJ-hTERT were a gift of Dr. D. Peeper (The Netherlands Cancer Institute, Amsterdam). IMR90-ER/RAS^V12 cells were a gift of Masashi Narita[25]. BJ fibroblasts were grown in high glucose Dulbecco's modified Eagle's medium with ultraglutamine (DMEM), supplemented with 10% tetracycline-free and heat-inactivated fetal bovine serum (FBS) and 100 U/mL penicillin/streptomycin (Lonza). hTERT-immortalized human retinal pigment epithelial RPE1-hTERT cells (CRL-4000, ATCC) were cultured in DMEM/F-12, supplemented with 10% heat-inactivated FBS and 100 U/mL penicillin/streptomycin. IMR90 fibroblasts were cultured in DMEM/F-12, supplemented with 10% heat-inactivated FBS and 100 U/mL penicillin/streptomycin. Cell lines were grown at 37 °C in a humidified atmosphere of 5% $CO_2$ and were tested for the absence of mycoplasma contamination.

### Construction and production of H-RAS^V12, TREX1 and TREX1-D18N lentiviral vectors
H-RAS^V12 from pBABE-Puro H-RAS^V12 (n°12545 Addgene) was cloned in a home-made inducible vector derived from pLVX-Tight-Puro (Clontech) to make it SIN by a deletion in the 3'-LTR. Specifically, the XhoI-KpnI fragment of pLVX-Tight-Puro containing the inducible promoter was subcloned in pHR-BX[64] and the H-RAS^V12 coding DNA sequence was introduced into the multiple cloning sites by appropriate restriction/ligation, as described previous[11]. GFP-TREX1 and GFP-TREX1 (D18N) plasmids, a gift from Judy Lieberman (Addgene plasmid 27219 and 27220[65]), were digested with AgeI and XbaI and the fusion genes subcloned with the appropriate adaptors in pHRTK[66], a home-made, cPPT-containing SIN lentiviral vector under the control of the constitutive CMV promoter or in the inducible pTripZ vector (Thermo Scientific) in place of the original TurboRFP. Virions were produced in HEK293T cells as previously described[67].

### Drugs and treatments
Doxycycline (Dox, purchased from SIGMA) was used at a final concentration 10 μg/ml (or otherwise stated in figure legends) to induce the RAS^V12 overexpression in BJ-hTert fibroblasts. Mirin, PFM01 and PFM39 (all from SIGMA) were dissolved in DMSO and used at a final concentration of 10 μM. The cGAS (RU.521) and STING (H-151) inhibitors were dissolved in DMSO and were purchased from Invivogen. According to drug treatments, the DMSO vehicle was used as a control. 4-hydroxytamoxifen (4-OHT, Sigma) was dissolved in ethanol and stored at −20 °C. Recombinant human interferon-β was purchased from PeproTech (reference 300-02BC), dissolved in water at a stock concentration of $2 \times 10^5$ Units per ml (U/ml) and stored in aliquots at −20 °C. Recombinant IFN-β was used at final concentrations ranging from 50 to 300 U/ml as previously described[6].

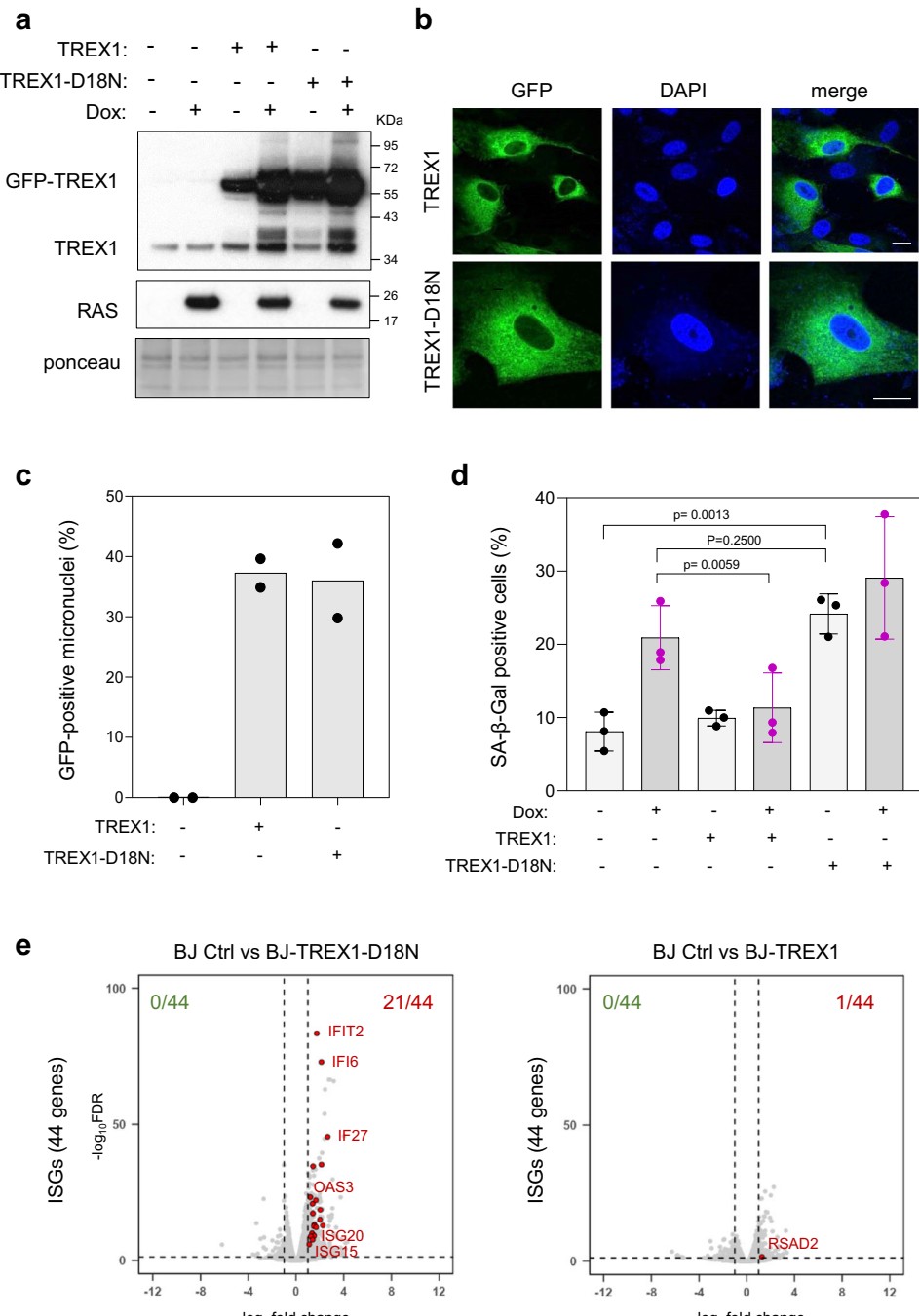

**Fig. 6 | TREX1 modulates senescence in BJ-RAS^V12 fibroblasts. a** Western blot analysis of RAS^V12 and TREX1 levels in BJ-RAS^V12 cells treated or not with 10 μg/ml doxycycline and stably overexpressing full-length (TREX1) or D18N mutant (TREX1-D18N) TREX1 tagged with an N-terminal GFP. Data are representative of at least three independent experiments. **b** Subcellular localization of GFP-TREX1 and GFP-TREX1-D18N was assessed by fluorescence microscopy. Representative images of one experiment are shown (n = 3). **c** Frequency of GFP-TREX1 positive micronuclei. At least 200 cells of each sample were scored for the formation of micronuclei (n = 2). **d** Frequency of SA-β-gal positive BJ-RAS^V12 fibroblasts overexpressing TREX1 or TREX1-D18N. Mean, SD and p values (two-sided unpaired t-test) are shown for three biological replicates. **e** Volcano plots of differentially expressed ISG genes in non-induced BJ-RAS cells (BJ Ctrl) compared to cells overexpressing either TREX1 or TREX1-D18N. Source data are provided as a Source Data file.

## Cell fractionation, Western blotting and antibodies

Cells were lysed in 2× Laemmli buffer at the concentration of $1 \times 10^4$ cells/μl. Lysates were treated with 3 μl of Benzonase (25 U/μl, Sigma) for 30 min at 37 °C. Cytosolic and nuclear fractions were obtained by lysing cells for 5 min on ice with a lysis buffer (10 mM Tris pH7.4, 10 mM NaCl, 10 mM MgCl$_2$) containing 0.25% NP-40. Nuclear pellets were lysed in 2x Laemmli buffer and digested with benzonase. Proteins were separated by sodium dodecyl sulfate polyacrylamide gel electrophoresis (SDS-PAGE), transferred to nitrocellulose membrane and analyzed by western immunoblotting with indicated antibodies: anti-RAS (1/500, BD, ref 610002), anti-MRE11 (1/500, Novus, NBS100-142), anti-cGAS (1/500, Cell Signaling, #15102), anti-p53 (1/500, Santa Cruz, clone DO7 sc-47698). Following antibodies were purchased from Abcam: anti-ATM (1:500, ab183324), anti-phospho S1981 ATM (1/500, ab81292), anti-IL1-a (1/500, ab9614), anti-ISG15 (1/1000, ab13346), anti-TREX1 (1:1000, ab185228). Rabbit anti-Phospho RPA32 (S33) and anti-

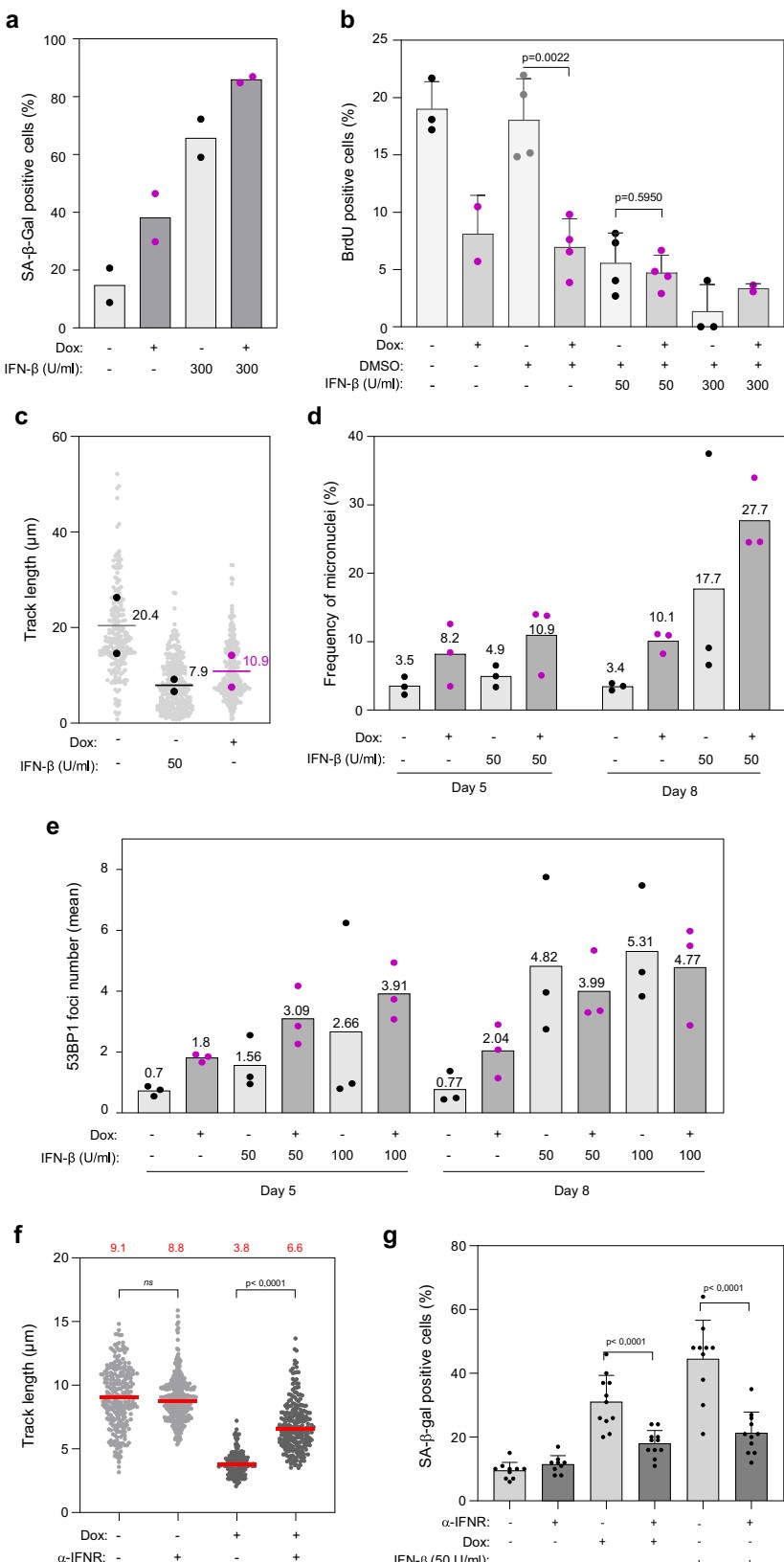

Phospho RPA32 (S4/S8) were from Bethyl (used at 1/500, A300-246A and A300-245A, respectively). Blots were incubated with horseradish peroxidase-linked secondary antibody (GE Healthcare) and visualized using either the ECL pico plus chemiluminescence (Pierce) or ECL Clarity Max (Biorad) and a chemidoc camera (Biorad).

## SA-β-galactosidase staining

Cells were cultured in the presence of the indicated drugs and the indicated times. Then were processed with commercial SA-β-galactosidase staining kits following provider instructions (Kit #9860 from Cell signaling).

**Fig. 7 | IFN-β induces replication stress and senescence in BJ fibroblasts. a** BJ-RAS[V12] fibroblasts induced or not with 10 μg/ml doxycycline were treated or not with 300 U/ml of recombinant interferon-β (IFN-β) for a period of 8 days and the frequency of SA-β-gal positive cells was scored in two independent experiments. **b** BrdU incorporation in BJ-RAS[V12] cells treated or not with 50 or 300 U/ml IFN-β. Mean, SD and p values (two-sided unpaired t-test) are shown. At least 90 cells were scored per condition and in each experiment. Each point represent a biological replicate. **c** DNA fiber analysis of fork progression in BJ-RAS[V12] cells treated or not with 10 μg/ml Dox and 50 U/ml IFN-β, as described in Fig. 4a. Median values from two independent experiments are indicated. **d** Micronuclei frequency in BJ-RAS[V12] cells treated or not with 10 μg/ml Dox and 50 μ/ml IFN-β for 5 and 8 days. Mean values are shown for three independent experiments. **e** Number of 53BP1 foci in BJ-RAS[V12] cells treated or not with Dox and with 50 or 100 U/ml IFN-β for 5 or 8 days. Mean values are shown for three independent experiments. **f** IFN-β signaling

contributes to RAS[V12]-induced fork slowing. BJ-RAS[V12] fibroblasts were treated or not with 10 μg/ml doxycycline and 50 U/ml IFN-β for 6 days as indicated, in the presence of an anti-IFNα/β receptor antibody (α-IFNR) or a control IgG. Replication fork progression was measured by DNA fiber spreading. A minimum of 150 fibers were measured in each sample. Median track length is indicated in red. ****p < 0.0001, ns non-significant, two-tailed Mann–Whitney rank sum test. A representative experiment is shown (n = 2). **g** IFN-β signaling promotes RAS[V12]-induced senescence. BJ-RAS[V12] fibroblasts treated as indicated above and β-galactosidase activity was analyzed on day 6. A minimum of 200 cells from 10 images were scored for SA-β-gal-positive cells in each sample. The percentage of SA-β-gal-positive cells (Mean ± SD) is shown for one representative experiment from two independent experiments (n = 2). ****p < 0.0001, two-tailed Mann–Whitney rank sum test. Source data are provided as a Source Data file.

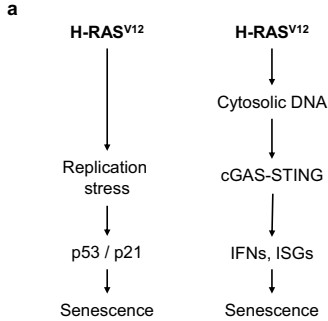

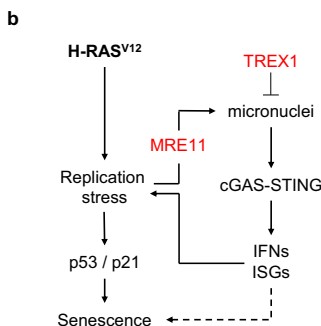

**Fig. 8 | Role of RS and cytosolic DNA sensing pathways in OIS. a** The DDR and cGAS-STING pathways are both required for the onset of OIS, but whether these pathways cooperate to induce senescence remains unclear. **b** Our data indicate that the nuclease MRE11 plays a central role in OIS by activating both the RS response and the cGAS-STING pathway via micronuclei formation. This function of MRE11 is counterbalanced by another nuclease, TREX1, which prevents activation of the cGAS-STING pathway by degrading cytosolic DNA. Remarkably, TREX1 inhibition or addition of IFN-β was sufficient to induce RS and senescence, even in the absence of RAS[V12] induction. Collectively, these data suggest that MRE11 activates a positive feedback loop that amplifies the RS response in OIS via activation of cytosolic DNA sensing.

### Growth curve determination

At day 0, $3 \times 10^5$ to $5 \times 10^5$ of cells were seeded in 25 cm² flasks or in 6-well plates with the indicated treatments. At each passage (p), each 3 to 4 days, cells were counted with the automated cell counter - countess II (ThermoFisher) and all re-platted at the same density.

### DNA fiber spreading

DNA fibers were obtained and analyzed as previously described[11,21]. Briefly, ongoing replication forks were sequentially pulse-labeled with thymidine analogues, 20 UM IdU and then 100 μM CldU, for 20 min each. Cells were harvested by trypsinization and lysed in spreading buffer (200 mM Tris-HCl pH 7.5, 50 mM EDTA, 0.5% SDS) on microscope slides. After fiber fixation in methanol/acetic acid (3:1 ratio) the

DNA fibers, and replication tracks were detected by immunostaining, using the following combination of antibodies. Primary antibody mix: Mouse anti-BrdU to detect IdU (1/200, BD 347580), Rat anti-BrdU to detect CldU (1/100, Eurobio clone BU1/75). Secondary antibody mix: Goat anti-rat Alexa 488 (1/200 PBS/T, Molecular Probes, A11006); Goat anti-mouse IgG1Alexa 546 (1/200 PBS/T, Molecular Probes, A21123). The ssDNA is detected with Anti-ssDNA (1/200, auto anti-ssDNA, DSHB) and then the goat anti-Mouse IgG2a Alexa 647 (1/200 PBS/T, Molecular Probes, A21241). Tracks were measured with FIJI. Statistical analysis of track lengths is performed with GraphPad Prism 9.0.

### DAPI staining to assess micronuclei frequency

To assess micronuclei formation, cells were grown on 12 mm diameter glass-coverslips (ref 0117520 Paul Marlenfeld GmbH & co), fixed 15–20 min in ice-cold 80% methanol, rinsed three times with PBS, and stained with a 1:5000 dilution of DAPI in PBS for 5–10 min at RT (DAPI was purchased from SIGMA). Alternatively, cells were fixed with 2% PFA 10 min at room temperature (RT) and then permeabilized with 0.1% Triton in PBS 10 min at RT. This last condition was used to estimate the frequency of cGAS positive micronuclei. Fixation was performed with 2% BSA in PBS 0.1% Triton for 45 min at RT. Immunostaining was performed overnight at 4 °C with rabbit anti-cGAS D1D3G (1:400, Cell Signaling, #15102) and secondary goat anti-rabbit coupled to Alexa 546 (1:500, Molecular Probes, A11035). To estimate TREX1 localization to micronuclei we took advantage of the fluorescence signal produced by the expression of the tagged GFP-TREX1 proteins. After DAPI staining, coverslips were washed three times with PBS and mounted with Pro-Long Gold Antifade (P36930, ThermoFischer Scientific). Images were acquired with a Zeiss Axioimager Apotome microscope.

### BrdU staining

To assess the proliferation of BJ fibroblasts, cells were grown on glass coverslips (as described above) and then were pulse-labeled for 20 min with 30 μM of BrdU, then rinsed with PBS and fixed in ice-cold 80% methanol (as described above). Coverslips were then rinsed three times with PBS and stored at 4 °C before immunostaining. DNA was denatured 45 min at RT in 2.5 M HCl, rinsed three times in PBS and then blocked 30–45 min in 2% BSA in PBS 0.1% Triton. For BrdU staining we used either a rat anti-BrdU (1:1000, Eurobio clone BU1/75) or mouse anti-BrdU (1:1000, BD 347580) antibodies 2 h at RT (1:1000, Eurobio clone BU1/75). After 3 washes, fluorescently-conjugated secondary antibodies were used at 1:500 final dilution. Cells were DAPI stained, coverslips mounted in Prolong Gold Antifade (P36930, ThermoFischer Scientific) and images acquired with a Zeiss Axioimager Apotome microscope.

### EdU Click chemistry and flow cytometry

BJ fibroblasts were pulse labeled with 10 μM EdU (5-ethynyl-2'-deoxyuridine) for 20 min after doxycycline induction and inhibitor

treatment as indicated. DNA synthesis was quantified using the Click-iT EdU flow cytometry assay kit (ThermoFisher Scientific, C10420) according to the manufacturer's instructions. Percentage of cells in S phase was quantified and analyzed using flow cytometry (MACSQuant analyser) and the software FlowJo.

## 53BP1 foci staining

Cells were grown on glass coverslips as described above, fixed with 2% PFA 10 min at RT and then permeabilized with 0.1% Triton in PBS 10 min at RT. Blocking was performed with 2% BSA in PBS 0.1% Triton for 45 min at RT. Immunostaining was performed overnight at 4 °C with rabbit anti-53BP1 antibody (1:2000, Novus, NB100-304). After three washes, fluorescently conjugated antibodies were used to detect the signals (1:1000).

## Comet assay

BJ fibroblasts were treated with doxycycline to induce TREX1 or TREX1-D18N for 8 days or with indicated concentrations of interferon-β for 6 days. Alkaline comet assay was performed using the OxiSelect Comet Assay Kit (Cell Biolabs, STA-351) according to manufacturer's instructions. Under alkaline conditions, this assay detects both ssDNA and dsDNA breaks. Twenty µM DNA topoisomerase II inhibitor, etoposide treatment for 5 h was included as a positive control.

## RNA-seq

RNA sequencing (RNA-seq) were performed as previously described[11,68]. Briefly, libraries were prepared using the Illumina TruSeq Stranded mRNA Library Prep Kit. Paired-end RNA-seq were performed with an Illumina NextSeq sequencing instrument (MGX-Biocampus, Montpellier, France). The expression level of genes was normalized to the library size using edgeR R/Bioconductor package. Differential expression analysis was performed using edgeR R/Bioconductor package. Genes with a FDR < 0.05 and a $\log_2$ fold change $< -1$ or $>1$ were considered as significantly differentially expressed. We focused our analysis on specific list of genes as follows. The hallmark of the Interferon-α response (IFN-α, from GSEA: M5911), Type I IFN-regulated genes with ISRE promoter elements (ISGs, from Reactome: R-HSA-9034125) and of SASP genes previously described[69].

## RT-qPCR

The expression of SASP markers IL6 and CXCL1 was quantified by qRT–PCR as described[21] and normalized to *GAPDH*. Reverse transcription was performed by using the Superscript III First Synthesis System for RT–PCR (ref. 18080-051, Invitrogen). A LightCycler 480 SYBR Green I Master Mix (ref. 04887352001, Invitrogen) was used to perform quantitative PCR.

## Statistical analysis

All figures and statistical analysis were performed with GraphPad Prism 9.0. P values of 0.05 or less were considered significant. The tests used are specified in the figure legends.

## Reporting summary

Further information on research design is available in the Nature Portfolio Reporting Summary linked to this article.

## Data availability

The data sets generated and/or analyzed during the current study are available from the corresponding authors on reasonable request. The RNA-seq datasets showed in Fig. 1 has been previously generated and reported[11], and are available in the GEO repository, accession number: GSE123380. All data generated in this study are provided in the Supplementary Information, Supplementary Data, and Source Data files Source data are provided with this paper.

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

## Acknowledgements

We thank Anne Letessier, Benjamin Pardo, Benoît Le Tallec and Benoît Miotto for their comments and suggestions on the manuscript. We thank the MRI imaging facility (Biocampus) for assistance with image acquisition and analysis. We thank Carl Mann (CEA/Saclay) for providing the IMR90-ER/RAS$^{V12}$ fibroblasts. H.T. has been supported by AIRC under the iCARE-2 fellowship program. Our project has received funding from AIRC and from the European Union's Horizon 2020 research and innovation program under the Marie Skłodowska-Curie grant agreement No 800924. Research carried by Y.L.L. is supported by the Fondation ARC pour la recherche sur le cancer (N°PJA 20191209522). Work in the Pasero lab is supported by grants from the Agence Nationale pour la Recherche (ANR), Institut National du Cancer (INCa), Ligue Nationale Contre le Cancer (équipe labellisée), Worldwide Cancer Research, Fondation ARC and Fondation MSDAVENIR.

## Author contributions

H.T. and Y.L.L. performed most of the experiments. H.T. wrote the original draft -edited and reviewed the manuscript and edited the figures. N.B., A.V., B.L., and D.G. provided technical assistance and contributed to image analysis. C.M. constructed the RAS$^{V12}$, TREX1, and TREX1-D18N lentiviral vectors. D.G., J.H., and J.M. performed RNA-seq bioinformatics analysis. Y.L.L. and P.P. conceived and supervised the study – edited and reviewed the manuscript and figures. All authors reviewed and accepted the manuscript before submission.

## Competing interests

The authors declare no competing interests.
