## [Peer Review File · Nature Communications]

MRE11 and TREX1 control senescence by coordinating replication stress and interferon signalingREVIEWER COMMENTS

Reviewer #1 (Remarks to the Author):

The study from Techer et al. aims at understanding the interplay between DNA replication stress and cGAS-STING pathway in oncogene-induced senescence (OIS), a major barrier to cell transformation. The work builds on previous findings from the same authors reporting that some clones of HRASV12 overexpressing fibroblasts with high levels of Claspin and Timeless, two critical regulators of fork stability, escape from senescence and restore normal fork progression, suggesting that RS promotes OIS. Similarly, cGAS/STING inhibitors by dampening interferon gene expression allow cells to escape from senescence. Next, the authors aimed at investigating whether there is a crosstalk between DNA replication stress and cGAS-STING mediated recognition of cytosolic DNA and inflammation or the two act as individual, mutually-exclusive, pathways. The paper is generally well-written and data are presented in a logic manner. The main findings are in line with previous literature. I have several comments/suggestions that may help the authors to strengthen their findings, especially to really elucidate the crosstalk between canonical RS response and cGAS-STING pathway at the onset of OIS.

FIGURE 1: In this figure the authors elegantly show that BJ-HRASV12 decreases fork rate, BrdU incorporation but increases IFN, ISGs and SASP genes expression, transcriptional responses that were mitigated in HRASV12 escaped clones from their previous paper (Bianco et al Nat Comm 2019). Interestingly, cGAS inhibitors treatment rescues BrdU incorporation in BJ-HRASV12, which is however not significant for STING inhibitors. These data are not very strong, and the effects are likely related to the dose of inhibitors employed. A dose response to both cGAS and STING inhibitors should be performed as well as a DNA fiber assay to elucidate the interplay between inflammation and RS as I would expect that these cGAS/STING inhibitors rescue DNA track lengths of RAS-expressing fibroblasts. It would be important also to measure SA-beta-GAL and DDR markers, together with proliferation/RS recovery, in this setting.

FIGURE 2: Experiment in panel A proves that Mirin treatment in BJ-HRASV12 significantly increases cell growth, even more compared to wild-type cells. Intriguingly, Mirin treatment affects senescence and restores proper DNA replication in these cells. Testing BrdU incorporation upon treatment with different inhibitors of either Mre11 endonuclease (PFM01) or exonuclease (PFM39) activity unveils that the exonuclease domain could be more important for OIS establishment although both functions seem to be involved in this process. Nevertheless, Mirin is inhibiting exonuclease activity but is not as efficient as PFM39: this point should be discussed or explored further by trying higher doses of Mirin in combination with DNA fiber assays experiments (also in the presence of PFM01 and PFM39). Levels of SA-beta-GAL in RAS expressing cells (control) are unusually low (15%). How many days after RAS expression were the cells analyzed? An experimental scheme depicting treatment doses and timing would help here.

FIGURE 3: Mirin rescues replication fork progression in BJ-HRASV12 measured as fiber length and fork symmetry and decreases ISG15 and IL1 α expression at the protein level. In addition, this effect is linked to a reduction in ATM phosphorylation, a marker of activated DDR, as well as RPA32 phosphorylation in Serine 33, which is ATR-dependent and one of the major marker of Replication Stress. The authors should discuss the possibility that these breaks originate from collapsed Replication forks as excessive Mre11 activity could result in inefficient HR-mediated restart or inefficient replication fork reversal in BJ HRASV12 cell line.

FIGURE 5: They investigated the impact of micronuclei on senescence upon overexpressing the major cytosolic nuclease TREX1 on OIS. For this purpose, they established a cell line constitutively expressing GFP-tagged version of either TREX1 WT or a dominant negative nuclease-dead (TREX1-D18N) protein. After assessing the correct localisation, they observe that the overexpression of this mutated protein is not able to diminish OIS in cells overexpressing H-RASV12 while TREX1 wild type rescues senescence, as expected. Accordingly, RNA-seq data show an increased expression of IFN genes in cells overexpressing TREX1-D18N in comparison to wild-type cells. This figure shows that cytosolic DNA plays a fundamental role in OIS triggering probably through IFN signalling, in line with previous literature. However, as this paper aims to discover a possible feedback loop between inflammation and DNA replication stress, it would be interesting to know if DNA replication patterns are perturbed upon TREX1 inhibition by measuring DNA track length and fork asymmetry in cells overexpressing TREX wild type vs the dominant negative form.

FIGURE 6: At this point they were interested in understanding the role of IFN signaling in senescence upon TREX1 inhibition. By treating BJ-HRASV12 cell line with two different doses of

IFN- β , they observe an increased percentage of senescent cells compared to the effect of RASV12 induction in correlation with a reduced BrdU incorporation. Is IFN- β driving hyper proliferation at early time points before inducing cell cycle arrest? They report a decreased rate of DNA replication upon either RASV12 induction or IFN β treatment, but they don't provide the data for the IFN β supply upon RASV12 induction where I would expect an additive effect. This experimental condition would help in the discussion of results illustrated in the panels D and E of this figure. It would also be important to understand how much of the phenotype (micronuclei, RS, DDR) in RS expressing cells depends on the IFN- β signaling. I recommend performing experiments with IFN Receptor blockades or genetic inactivation to address this issue. The authors also observe an increased percentage of micronuclei and 53BP1 foci in cells treated with IFN- β and overexpressing the RASV12 oncogene, which is slightly higher compared to IFN- β or RASV12 overexpression alone DAY 8 suggesting an additive effect. Altogether, these data highlight that IFN- β treatment or overexpression of a dominant-negative mutant of TREX1 is sufficient to induce RS-mediated senescence resulting in the accumulation of micronuclei and activation of the cGAS-STING pathway. The authors claim that there may be DSB accumulation upon the overexpression of the dominant negative form of TREX1 but no evidence to support this hypothesis. A comet assay experiment should be included to strengthen this conclusion. In addition, it would be worthwhile assessing the presence of aneuploidy and lagging chromosomes in this context.

Minor comments:

- There is a missing connection between the first paragraph and the second paragraph in page 4 when they switch from cGAS-STING to inflammation. Please elaborate.
- Please double-check reference formatting in the text.
- Statistical analyses (and inferred p-values) should never be performed with $n \leq 5$ biological replicates.

Reviewer #2 (Remarks to the Author):

In the present manuscript the authors explore the connection between the cGAS-STING pathway and DNA damage response (DDR) in the induction of Oncogene Induced Senescence (OIS). Experimentally, they rely on a previously established immortalized cellular system consisting of human BJ fibroblasts overexpressing H-RASV12 (causing replication stress) to study this interplay in OIS establishment. In this cellular context, the authors show that MRE11 contributes to OIS by slowing down replication forks. Also, MRE11 facilitates the formation of micronuclei and activation of DDR, while it also correlates with the induction of IFN, SASP and ISG genes. Moreover, the authors demonstrate the cytoplasmic endonuclease TREX1 negatively regulates senescence through the degradation of cytosolic DNA, while IFN- β treatment (without overexpression of RAS) is sufficient to induce replication stress and senescence. The manuscript is well written and easy to follow and could contribute to the field of OIS. Nevertheless, several points require clarifications in order to be published.

Major comments

1. All experiments in the manuscript were performed in BJ fibroblasts. It would be critical if some of the key experiments are recapitulated in a second cellular system, ideally epithelial, considering that generally human tumors are predominantly of epithelial origin. This would strengthen the proposed concept.
2. Considering the critical role examined for MRE11, why did the authors use only the Mirin inhibitor to silence it and not also specific siMRE11 knockdown? Moreover, could overexpression of MRE11 lead to similar results as Claspin and Timeless?
3. The authors perform Western blot for phospho-ATM (S1981) and phospho-RPA32 (S33) to demonstrate the occurrence of DNA damage response. Total levels of ATM and RPA32 should be included in the immunoblot.
4. Since MRE11 is part of the MRN complex, what is the status and role of the other two subunits of the complex (RAD50 and NBS1)? Are they also implicated in the proposed hypothesis or the role of MRE11 is independent of the other factors?
5. In Figure 6, 53BP1 foci should be accompanied by other DNA damage markers/assays such as

γ H2Ax and comet assay.

6. Provided that appropriate antibodies are available to examine both MRE11 and TREX1 in tumors from patients or omics data on the expression status of these factors, I would suggest to look into the outcome of cancer patients stratified according to the expression status of these factors.

Minor comments

1. Why certain experiments are confirmed only in duplicates and others in triplicates? There should be a consistency.
2. Please provide quantification for the protein levels in Figure 3d.
3. I may have missed it, but why in Figure 5a induction of RAS was performed with 2 μ g/ml of Dox. while in the rest of the experiments with 10 μ g/ml?
4. Define scale bars of Figure 5b
5. Define scale bar of Figure Supp. 3b
6. Provide statistic analysis for Figure 6e
7. Some of the representative images from Figure Supp 2a could be moved into Figure 2b to support it.
8. Some of the representative images from Figure Supp 3b could be moved into Figure 3c to support it.

Reviewer #1 (Remarks to the Author):

The study from Techer et al. aims at understanding the interplay between DNA replication stress and cGAS-STING pathway in oncogene-induced senescence (OIS), a major barrier to cell transformation. The work builds on previous findings from the same authors reporting that some clones of HRASV12 overexpressing fibroblasts with high levels of Claspin and Timeless, two critical regulators of fork stability, escape from senescence and restore normal fork progression, suggesting that RS promotes OIS. Similarly, cGAS/STING inhibitors by dampening interferon gene expression allow cells to escape from senescence. Next, the authors aimed at investigating whether there is a crosstalk between DNA replication stress and cGAS-STING mediated recognition of cytosolic DNA and inflammation or the two act as individual, mutually-exclusive, pathways.

The paper is generally well-written and data are presented in a logic manner. The main findings are in line with previous literature. I have several comments/suggestions that may help the authors to strengthen their findings, especially to really elucidate the crosstalk between canonical RS response and cGAS-STING pathway at the onset of OIS.

We thank Reviewer #1 for his/her positive comments and suggestions. We have carried out most of the experiments he/she suggested to strengthen our case.

FIGURE 1: In this figure the authors elegantly show that BJ-HRASV12 decreases fork rate, BrdU incorporation but increases IFN, ISGs and SASP genes expression, transcriptional responses that were mitigated in HRASV12 escaped clones from their previous paper (Bianco et al Nat Comm 2019). Interestingly, cGAS inhibitors treatment rescues BrdU incorporation in BJ-HRASV12, which is however not significant for STING inhibitors. These data are not very strong, and the effects are likely related to the dose of inhibitors employed. A dose response to both cGAS and STING inhibitors should be performed as well as a DNA fiber assay to elucidate the interplay between inflammation and RS as I would expect that these cGAS/STING inhibitors rescue DNA track lengths of RAS-expressing fibroblasts. It would be important also to measure SA-beta-GAL and DDR markers, together with proliferation/RS recovery, in this setting.

We agree with this Reviewer that our experiments using cGAS and STING inhibitors were not entirely conclusive. We have now repeated DNA fiber experiments in BJ-RAS^{V12} cells with increasing concentrations of the cGAS (RU.521; 5, 10 and 20 μ M) and STING (H-151; 0.5, 1 and 5 μ M) inhibitors. The new data shown in **Figure 2b** confirm that both cGAS and STING inhibitors at least partially rescue the fork slowing mediated by RAS^{V12}. Of note, we observed that the cGAS inhibitor induced fork slowdown in the absence of RAS^{V12}, when used at 20 μ M, suggesting that such a high concentration may have deleterious effects on DNA replication.

We have also quantified the effect of cGAS and STING inhibitors on DNA synthesis by flow cytometry using EdU click chemistry. The new experiment shown in **Supplementary Fig. 2b** also indicates that both cGAS and STING inhibitors suppress RAS^{V12}-mediated replication inhibition to the same extent.

Finally, we have tested the ability of different concentrations of cGAS and STING inhibitors to prevent the formation of senescence-associated heterochromatin foci (SAHF) in IMR90 human lung fibroblasts expressing H-RAS^{V12} fused to the estrogen receptor (IMR90/ER-RAS^{V12}). Again, cGAS and STING inhibitors were able to revert the effect of the oncogenic stress in a dose-dependent manner. These

data are shown in a new **Figure 2c**. Together, these results support the view that the cGAS-STING pathway contributes to senescence induced by oncogenic stress.

FIGURE 2: Experiment in panel A proves that Mirin treatment in BJ-HRASV12 significantly increases cell growth, even more compared to wild-type cells. Intriguingly, Mirin treatment affects senescence and restores proper DNA replication in these cells. Testing BrdU incorporation upon treatment with different inhibitors of either Mre11 endonuclease (PFM01) or exonuclease (PFM39) activity unveils that the exonuclease domain could be more important for OIS establishment although both functions seem to be involved in this process. Nevertheless, Mirin is inhibiting exonuclease activity but is not as efficient as PFM39: this point should be discussed or explored further by trying higher doses of Mirin in combination with DNA fiber assays experiments (also in the presence of PFM01 and PFM39).

As requested by this Reviewer, we repeated DNA fiber spreading experiments in BJ-RAS^{V12} cells using higher doses of Mirin. Since 50 μ M Mirin was toxic to BJ cells, we compared the effect on fork rate of 10 and 20 μ M Mirin, 10 μ M PFM01 and 10 μ M PFM39. We found that although MRE11 inhibitors had little to no effect on fork speed in the absence of oncogenic stress, both concentrations of Mirin were able to restore normal fork progression in cells overexpressing RAS^{V12}. Moreover, these experiments confirmed that PFM01, but not PFM39, efficiently rescues fork slowdown in BJ-RAS^{V12} cells. These results are consistent with the fact that PFM01, but not PFM39, restores cell proliferation in BJ-RAS^{V12} cells (Fig. 3d). They also support the view that the endonuclease activity of MRE11 mediates oncogene-induced replication stress. These new data are displayed in a new **Figure 4c**.

We have now extended the discussion to the possible involvement of both the endo- and exonuclease activity of MRE11 in genome stability, affecting the RS response and senescence. We also mention that Mirin was initially characterized as an inhibitor of ATM signaling (PMID: 18176557), which may explain the observed difference between Mirin and PFM39 in our experimental setting.

To confirm the effect of Mirin in mitigating RAS^{V12}- induced fork slowing in a different cellular model, we analyzed fork speed in IMR90 fibroblasts overexpressing RAS^{V12}. The results shown in a new **Supplementary Fig. 4b** indicate that 20 μ M Mirin almost completely restored normal fork speed in the presence of RAS^{V12}. In addition, we now show in a new **Supplementary Fig. 3b** that Mirin prevents the formation of SAHF in IMR90/ER-RAS^{V12} cells, in a dose dependent manner.

Levels of SA-beta-GAL in RAS expressing cells (control) are unusually low (15%). How many days after RAS expression were the cells analyzed? An experimental scheme depicting treatment doses and timing would help here.

SA- β -gal detection was mostly carried out between day 6 and 14 following RAS^{V12} induction in BJ fibroblasts. This information is now better explained in the figure legend. We agree that RAS^{V12}-induced SA- β -gal was relatively low in these cells, especially when compared to the effect of IFN- β treatment (new **Figure 7a**). Therefore, we have used SAHF in IMR90-ER/RAS^{V12} cells (**Supplementary Fig. 1e, 3b**) as an additional senescence marker to strengthen our point.

FIGURE 3: Mirin rescues replication fork progression in BJ-HRASV12 measured as fiber length and fork symmetry and decreases ISG15 and IL1 α expression at the protein level. In addition, this effect is

linked to a reduction in ATM phosphorylation, a marker of activated DDR, as well as RPA32 phosphorylation in Serine 33, which is ATR-dependent and one of the major marker of Replication Stress. The authors should discuss the possibility that these breaks originate from collapsed Replication forks as excessive Mre11 activity could result in inefficient HR-mediated restart or inefficient replication fork reversal in BJ HRASV12 cell line.

This is indeed an interesting possibility that is now discussed in the manuscript.

FIGURE 5: They investigated the impact of micronuclei on senescence upon overexpressing the major cytosolic nuclease TREX1 on OIS. For this purpose, they established a cell line constitutively expressing GFP-tagged version of either TREX1 WT or a dominant negative nuclease-dead (TREX1-D18N) protein. After assessing the correct localisation, they observe that the overexpression of this mutated protein is not able to diminish OIS in cells overexpressing H-RASV12 while TREX1 wild type rescues senescence, as expected. Accordingly, RNA-seq data show an increased expression of IFN genes in cells overexpressing TREX1-D18N in comparison to wild-type cells. This figure shows that cytosolic DNA plays a fundamental role in OIS triggering probably through IFN signalling, in line with previous literature. However, as this paper aims to discover a possible feedback loop between inflammation and DNA replication stress, it would be interesting to know if DNA replication patterns are perturbed upon TREX1 inhibition by measuring DNA track length and fork asymmetry in cells overexpressing TREX wild type vs the dominant negative form.

We thank Reviewer #1 for raising this interesting question. We have now performed DNA fiber experiments in BJ fibroblasts to address this possibility. As shown in a new **Supplementary Fig. 6e**, the overexpression of both isoforms induced a slight reduction in fork speed that was not statistically significant for the mutant form. We also confirmed these results in RPE1-hTERT cells expressing doxycycline-inducible TREX1 or TREX1-D18N (**Panel 1 for reviewers**). Despite the fact that TREX1-D18N by itself induces senescence, no significant change in replication fork progression was observed. These data suggest that the effect of TREX1 and TREX1-D18N on fork progression in BJ-RAS^{V12} cells is unlikely to reflect a direct effect of these factors on replication forks. They also suggest that TREX1-D18N induces senescence through a mechanism distinct from that of RAS^{V12}.

Panel 1: Overexpression of TREX1 or TREX1(D18N) does not affect replication fork progression. hTERT-immortalized RPE1 cells expressing tetracycline-inducible TREX1 or the dominant negative mutant D18N were treated or not with 10 µg/ml doxycycline for 6 days. Replication fork progression was measured by DNA fiber spreading as described in the Materials and Methods. The median IdU+CldU track length is indicated in red.

FIGURE 6: At this point they were interested in understanding the role of IFN signaling in senescence upon TREX1 inhibition. By treating BJ-HRASV12 cell line with two different doses of IFN-β, they observe an increased percentage of senescent cells compared to the effect of RASV12 induction in

correlation with a reduced BrdU incorporation. Is IFN- β driving hyper proliferation at early time points before inducing cell cycle arrest?

To address this possibility, proliferation of BJ fibroblasts was monitored every 24 hours using the WST-1 assay. When compared to untreated BJ fibroblasts, we detected no increased proliferation at early time points following IFN- β treatment, until day 5 when the inhibitory effect of IFN- β appeared (**Panel 2 for reviewers**). From this respect, the effect of IFN- β differs from the one of RAS^{V12}.

Panel 2: Interferon- β does not affect cell proliferation at the early stage of treatment. BJ fibroblasts were treated with increasing doses of interferon β . The cell proliferation was quantified daily by WST-1 assay. Absolute optical absorbance at 450 nm was calculated according to the manufacturer's instructions.

They report a decreased rate of DNA replication upon either RASV12 induction or IFN β treatment, but they don't provide the data for the IFN β supply upon RASV12 induction where I would expect an additive effect. This experimental condition would help in the discussion of results illustrated in the panels D and E of this figure.

To determine whether there is an additive effect on DNA replication between RAS^{V12} induction and IFN- β treatment, we conducted DNA fiber spreading assay 6 days after doxycycline and IFN- β treatment. We found that there is weak but significant additive effect on fork slowdown between RAS^{V12} and IFN- β . The results are shown in **Supplementary Fig. 7h**. In parallel, cells were analyzed with SA- β -galactosidase staining (**Supplementary Fig. 7i**) and showed again a slight additive effect between RAS^{V12} and IFN- β .

It would also important to understand how much of the phenotype (micronuclei, RS, DDR) in RS expressing cells depends on the IFN- β signaling. I recommend performing experiments with IFN Receptor blockades or genetic inactivation to address this issue.

We thank Reviewer #1 for suggesting this interesting experiment. To assess the contribution of IFN- β signaling in RAS-induced senescence, we analyzed fork progression in BJ-RAS^{V12} cells in the presence of an anti-IFN α/β receptor (α -IFNR) antibody or of an isotypic IgG. Our data show that the presence of α -IFNR antibody suppressed 40 to 50% of the RAS-induced fork slowdown (**Figure 7f**). A similar suppression was observed with SA- β -gal staining (**Figure 7g**), suggesting that IFN- β signaling contributes significantly to RAS-induced senescence in the absence of exogenous addition of IFN- β .

The authors also observe an increased percentage of micronuclei and 53BP1 foci in cells treated with IFN- β and overexpressing the RASV12 oncogene, which is slightly higher compared to IFN- β or RASV12 overexpression alone DAY 8 suggesting an additive effect. Altogether, these data highlight that IFN- β treatment or overexpression of a dominant-negative mutant of TREX1 is sufficient to

induce RS-mediated senescence resulting in the accumulation of micronuclei and activation of the cGAS-STING pathway. The authors claim that there may be DSB accumulation upon the overexpression of the dominant negative form of TREX1 but no evidence to support this hypothesis. A comet assay experiment should be included to strengthen this conclusion. In addition, it would be worthwhile assessing the presence of aneuploidy and lagging chromosomes in this context.

We agree with Reviewer #1 that we did not provide direct evidence for the occurrence of DSBs, besides 53BP1 foci, which is a classical but indirect DSB marker. As suggested by this reviewer, we have now performed comet assay. These experiments show that although the overexpression of TREX1 and TREX1-D18N was not sufficient to induce detectable levels of DNA damage (Supplementary Fig. 6f), IFN- β treatment was sufficient to induce DNA breaks in a dose-dependent manner (Supplementary Fig. 7g), even in the absence of oncogenic stress. Since comet assays were performed in alkaline conditions, these results indicate the presence of both ssDNA and dsDNA breaks, and not specifically of DSBs. This is now indicated in the text.

Minor comments:

- There is a missing connection between the first paragraph and the second paragraph in page 4 when they switch from cGAS-STING to inflammation. Please elaborate.

We have now clarified this transition.

- Please double-check reference formatting in the text.

Sorry for the formatting errors. They have been corrected.

- Statistical analyses (and inferred p-values) should never be performed with n or equal 5 biological replicates.

We agree with Reviewer #1 that in principle, statistical analyses (and inferred p-values) should not be performed with n or equal 5 biological replicates. Due to time constraints, we have decided to consolidate our findings by extending our study to additional cellular models, such as IRM90 fibroblasts and RPE-1 epithelial cells. Large datasets (more than 150 fibers or 200 cells) are presented as superplots of biological replicates or as individual experiments with a rank sum test of the distributions. We have retained statistical analyses for experiments with $n=3$ or more, but we are happy to remove these analyses if preferred.

Reviewer #2 (Remarks to the Author):

In the present manuscript the authors explore the connection between the cGAS-STING pathway and DNA damage response (DDR) in the induction of Oncogene Induced Senescence (OIS). Experimentally, they rely on a previously established immortalized cellular system consisting of human BJ fibroblasts overexpressing H-RASV12 (causing replication stress) to study this interplay in OIS establishment. In this cellular context, the authors show that MRE11 contributes to OIS by slowing down replication forks. Also, MRE11 facilitates the formation of micronuclei and activation of DDR, while it also correlates with the induction of IFN, SASP and ISG genes. Moreover, the authors

demonstrate the cytoplasmic endonuclease TREX1 negatively regulates senescence through the degradation of cytosolic DNA, while IFN- β treatment (without overexpression of RAS) is sufficient to induce replication stress and senescence. The manuscript is well written and easy to follow and could contribute to the field of OIS. Nevertheless, several points require clarifications in order to be published.

We thank Reviewer #2 for positive comments. We hope that we have properly addressed her/his concerns in this revised version of our manuscript.

Major comments

1. All experiments in the manuscript were performed in BJ fibroblasts. It would be critical if some of the key experiments are recapitulated in a second cellular system, ideally epithelial, considering that generally human tumors are predominantly of epithelial origin. This would strengthen the proposed concept.

We agree that it was important to extend our main findings to other cell lines. We have now reproduced all key experiments in two other immortalized human cell lines, IMR90 lung fibroblast and RPE1 retinal pigment epithelial cells:

- We first recapitulated H-RAS^{V12}-induced senescence in IMR90-ER/RAS^{V12} cells (**Supplementary Fig. 1a**), using senescence markers such as fork slowdown (**Supplementary Fig. 1b**) replication inhibition (**Supplementary Fig. 1c and 1d**) and SAHF (**Supplementary Fig. 1e**).
- We show that Mirin rescues fork slowing (**Supplementary Fig. 4b**) and inhibits SAHF (**Supplementary Fig. 3b**) in IMR90-ER/RAS^{V12} cells. SAHF inhibition was also observed in the presence of cGAS and STING inhibitors (**Fig. 2c**).
- We reproduced the effect of IFN- β on EdU incorporation and fork speed in IMR90-ER/RAS^{V12} cells (**Supplementary Fig. 7a and 7b**).
- We confirmed that IFN- β treatment caused growth inhibition (**Supplementary Fig. 7c**), fork slowdown (**Supplementary Fig. 7d**), micronuclei formation (**Supplementary Fig. 7e**) and DNA damage (**Supplementary Fig. 7g**) in RPE1-hTERT cells.

Together, these results indicate that the phenotypes observed in IMR90 and RPE-1 cells fully recapitulate those observed in BJ fibroblasts.

2. Considering the critical role examined for MRE11, why did the authors use only the Mirin inhibitor to silence it and not also specific siMRE11 knockdown? Moreover, could overexpression of MRE11 lead to similar results as Claspin and Timeless?

We have tried to deplete MRE11 with siRNAs and shRNAs in BJ-RAS^{V12} cells, but since all subunits of the MRE11 complex are essential for viability in proliferating cells (PMID: 21252998), we have been unable to monitor cell growth and fork progression in MRE11-KD cells. In principle, the overexpression of MRE11 should lead to similar results as Claspin and Timeless. We have not addressed this possibility experimentally as it would require the concomitant overexpression of RAD50 and NBS1. Yet, we have previously reported that MRE11A is overexpressed in BJ-RAS^{V12} clones adapting to oncogene-induced RS (PMID: 30796221).

3. The authors perform Western blot for phospho-ATM (S1981) and phospho-RPA32 (S33) to demonstrate the occurrence of DNA damage response. Total levels of ATM and RPA32 should be included in the immunoblot.

Total ATM levels and Western blot quantifications are now included.

4. Since MRE11 is part of the MRN complex, what is the status and role of the other two subunits of the complex (RAD50 and NBS1)? Are they also implicated in the proposed hypothesis or the role of MRE11 is independent of the other factors?

As mentioned above, all subunits of the MRE11 complex are essential for the viability of proliferating cells. Since MRE11 is the only subunit that can be targeted with chemical inhibitors, we have not been able to address this question experimentally. However, it should be noted that all the known functions of MRE11 in DNA repair and cGAS signaling depend on the three subunits of the complex, so it is very likely that this is also the case for the role of MRE11 in OIS.

5. In Figure 6, 53BP1 foci should be accompanied by other DNA damage markers/assays such as γ H2Ax and comet assay.

As indicated in our response to Reviewer #1, we have used comet assay to provide evidence that IFN- β induces DNA breaks in a dose-dependent manner (**Supplementary Fig. 7g**), which is consistent with increased 53BP1 foci (**Fig. 7e**).

6. Provided that appropriate antibodies are available to examine both MRE11 and TREX1 in tumors from patients or omics data on the expression status of these factors, I would suggest to look into the outcome of cancer patients stratified according to the expression status of these factors.

We have analyzed omics data for different cancers and found correlations between MRE11 expression and overall survival, as illustrated below for liver cancer patients (**Panel 3 for reviewers**). In these tumors, the MRE11A gene is significantly overexpressed relative to normal tissues, unlike TREX1. Kaplan-Meier curves indicate that this overexpression is associated with a poor prognosis, which seems to argue against a role in OIS. However, MRE11 has pleiotropic functions in DNA repair and that its increased expression could reflect its role in the adaptation to replication stress, as reported earlier for Claspin and Timeless (PMID: 30796221). As indicated above, MRE11A is overexpressed in B1-RAS^{V12} clones escaping oncogene-induced RS (PMID: 30796221). However, we do not know if it is also the case in tumor samples, as MRE11A was not included in the original panel of genes that were analyzed in tumor samples and adjacent normal tissues and normalized for cell proliferation (PMID: 23552402). Considering all these caveats, we have decided not to include cancer data in the manuscript.

Panel 3 : MRE11 and TREX1 expression was assessed in liver cancer using TNM-plot (<https://tnmplot.com>) using gene-chip data and paired tumor and normal tissues for comparison. The overall survival of liver cancer patients was assessed using Kaplan-Meier Plotter online tool (<https://kmplot.com>). Patients were separated using the auto cutoff parameter.

Minor comments

1. Why certain experiments are confirmed only in duplicates and others in triplicates? There should be a consistency.

All the key experiments were performed at least in triplicates. A limited number of confirmatory experiments are shown in duplicates. Considering time constraints, we favored to confirm our main observations in different cell lines (IMR90 fibroblasts and RPE-1 epithelial cells).

2. Please provide quantification for the protein levels in Figure 3d.

The quantification is included now.

3. I may have missed it, but why in Figure 5a induction of RAS was performed with 2µg/ml of Dox. while in the rest of the experiments with 10µg/ml?

Sorry for the confusion, all RAS inductions were performed with 10 µg/ml Dox.

4. Define scale bars of Figure 5b

Scale bar is now defined.

5. Define scale bar of Figure Supp. 3b

Scale bar is now defined.

6. Provide statistic analysis for Figure 6e

The graph and statistical analysis are now shown in Supplementary Fig. 6h.

7. Some of the representative images from Figure Supp 2a could be moved into Figure 2b to support it.

We have moved representative images of SA- β -gal staining from day 14 to the main figure (now **Fig. 3b**)

8. Some of the representative images from Figure Supp 3b could be moved into Figure 3c to support it.

We have moved representative images of micronuclei to the main figure (now **Fig. 4e**).

REVIEWERS' COMMENTS

Reviewer #1 (Remarks to the Author):

The study conducted by Techer et al. seeks to elucidate the relationship between DNA replication stress and the cGAS-STING pathway in oncogene-induced senescence (OIS), a crucial defence mechanism against cell transformation. Building upon their prior research, the authors observed that certain clones of H-RASV12-overexpressing fibroblasts, characterized by elevated levels of Claspin and Timeless proteins (key regulators of fork stability—demonstrated the ability to evade senescence and restore normal fork progression. This suggests that replication stress (RS) plays a role in promoting OIS. Additionally, the inhibition of cGAS/STING activity, which suppresses interferon gene expression, enabled cells to bypass senescence. In this manuscript, the authors aimed to explore there is an interplay between DNA replication stress and cGAS-STING-mediated recognition of cytosolic DNA leading to inflammation, or if these pathways operate independently. The authors clearly demonstrate that the nucleases MRE11 and TREX1 regulated OIS in different immortalized human cell lines overexpressing H-RASV12, providing a mechanistic link between Oncogene-induced RS and IFN signalling in OIS.

The revised version of this paper answered to the main issues raised in the previous review. After this round of revision, this manuscript is even better written and contains all the improvements suggested by reviewers, consolidating the main findings. Therefore, now this paper is suitable for publication in this journal. However, this manuscript requires a few extra modifications prior to publication, which are not essential for its acceptance and would further ameliorate the clarity of the paper: -this paper lacks a paragraph about future perspectives. Few lines about potential applications in cancer therapy and/or future experiments should be added in the discussion session; -quantification of comet assays in Supplementary Figure 6f and Supplementary Figure 7g. I suspect that there may be a slight effect on DNA damage upon TREX1 and TREX1-D18N overexpression; -the authors conclude from data in Figure 6 that TREX1 and TREX1-D18N trigger senescence through a mechanism distinct from that of RASV12. For this reason, it would be interesting to know in the future perspectives of the manuscript whether the authors have an idea of how mutated TREX1 induces senescence even in absence of H-RASV12 overexpression (e.g. starting from analysis of RNA-seq data); -lines 19-20 of the results sections should be modified: "We recently showed that a small fraction of BJ cells maintained under long-term RASV12 induction and eventually escaped senescence".

Reviewer #2 (Remarks to the Author):

The authors have properly addressed all issues raised and therefore I recommend publication.

Reviewer #1 (Remarks to the Author):

The study conducted by Techer et al. seeks to elucidate the relationship between DNA replication stress and the cGAS-STING pathway in oncogene-induced senescence (OIS), a crucial defence mechanism against cell transformation. Building upon their prior research, the authors observed that certain clones of H-RASV12-overexpressing fibroblasts, characterized by elevated levels of Claspin and Timeless proteins (key regulators of fork stability—demonstrated the ability to evade senescence and restore normal fork progression. This suggests that replication stress (RS) plays a role in promoting OIS. Additionally, the inhibition of cGAS/STING activity, which suppresses interferon gene expression, enabled cells to bypass senescence. In this manuscript, the authors aimed to explore there is an interplay between DNA replication stress and cGAS-STING-mediated recognition of cytosolic DNA leading to inflammation, or if these pathways operate independently. The authors clearly demonstrate that the nucleases MRE11 and TREX1 regulated OIS in different immortalized human cell lines overexpressing H-RASV12, providing a mechanistic link between Oncogene-induced RS and IFN signalling in OIS.

The revised version of this paper answered to the main issues raised in the previous review. After this round of revision, this manuscript is even better written and contains all the improvements suggested by reviewers, consolidating the main findings. Therefore, now this paper is suitable for publication in this journal. However, this manuscript requires a few extra modifications prior to publication, which are not essential for its acceptance and would further ameliorate the clarity of the paper:

We thank Reviewer #1 for his/her very positive comments on our revised manuscript.

-this paper lacks a paragraph about future perspectives. Few lines about potential applications in cancer therapy and/or future experiments should be added in the discussion session;

We have added the following paragraph at the end of the Discussion section: “Our findings raise important questions regarding the mechanisms by which MRE11, TREX1-D18N and IFN- β induce RS and on the potential role of downstream factors such as ISG15 in this process. Further work is also required to demonstrate that inflammation-mediated RS is necessary and sufficient to recapitulate the effect of RASV12 in vivo. Addressing these issues will allow us to better understand the interplay between replication stress and inflammation during oncogene-induced senescence and how this important defense mechanism protects complex organisms from cell transformation.”

-quantification of comet assays in Supplementary Figure 6f and Supplementary Figure 7g. I suspect that there may be a slight effect on DNA damage upon TREX1 and TREX1-D18N overexpression;

We have now included the quantification of the comet assay in Supplementary Figure 6f (7g was already in the revised manuscript). There is indeed a slight effect of DNA damage upon TREX1 overexpression and a much stronger effect upon TREX1-D18N expression.

-the authors conclude from data in Figure 6 that TREX1 and TREX1-D18N trigger senescence through a mechanism distinct from that of RASV12. For this reason, it would be interesting to know in the future

perspectives of the manuscript whether the authors have an idea of how mutated TREX1 induces senescence even in absence of H-RASV12 overexpression (e.g. starting from analysis of RNA-seq data);

We have tried to follow this advice and compared the list of differentially expressed genes in cells overexpressing RASV12 and TREX1-D18N, but the very large number of deregulated genes did not reveal any interpretable pattern. Further work is therefore required to address this important question.

-lines 19-20 of the results sections should be modified: “We recently showed that a small fraction of BJ cells maintained under long-term RASV12 induction and eventually escaped senescence”.

Corrected, thanks for spotting this out.

Reviewer #2 (Remarks to the Author):

The authors have properly addressed all issues raised and therefore I recommend publication.

We would like to thank Reviewer #2 for his/her very positive assessment of our work.